# Integrative omics analysis of the termite gut system adaptation to *Miscanthus* diet identifies lignocellulose degradation enzymes

Magdalena Calusinska [1✉], Martyna Marynowska[1,2], Marie Bertucci[1,3], Boris Untereiner[1], Dominika Klimek[1], Xavier Goux[1], David Sillam-Dussès[4], Piotr Gawron[5], Rashi Halder [5], Paul Wilmes[5], Pau Ferrer [1,6], Patrick Gerin [3], Yves Roisin[2] & Philippe Delfosse[1,7]

*Miscanthus* sp. biomass could satisfy future biorefinery value chains. However, its use is largely untapped due to high recalcitrance. The termite and its gut microbiome are considered the most efficient lignocellulose degrading system in nature. Here, we investigate at holobiont level the dynamic adaptation of *Cortaritermes* sp. to imposed *Miscanthus* diet, with a long-term objective of overcoming lignocellulose recalcitrance. We use an integrative omics approach combined with enzymatic characterisation of carbohydrate active enzymes from termite gut Fibrobacteres and Spirochaetae. Modified gene expression profiles of gut bacteria suggest a shift towards utilisation of cellulose and arabinoxylan, two main components of *Miscanthus* lignocellulose. Low identity of reconstructed microbial genomes to closely related species supports the hypothesis of a strong phylogenetic relationship between host and its gut microbiome. This study provides a framework for better understanding the complex lignocellulose degradation by the higher termite gut system and paves a road towards its future bioprospecting.

[1] BioSystems and Bioprocessing Engineering, Luxembourg Institute of Science and Technology, Rue du Brill 41, L-4422 Belvaux, Luxembourg. [2] Evolutionary Biology and Ecology, Université Libre de Bruxelles, Avenue F.D. Roosevelt 50, 1050 Brussels, Belgium. [3] Laboratory of Bioengineering, Earth and Life Institute, Université Catholique de Louvain, Croix du Sud 2/L7.05.19, 1348 Louvain-la-Neuve, Belgium. [4] Laboratory of Experimental and Comparative Ethology EA444, University Paris 13-Sorbonne Paris Cité, 399 Avenue Jean-Baptiste Clément, 93430 Villetaneuse, France. [5] Luxembourg Centre for Systems Biomedicine, University of Luxembourg, 7 Avenue des Hauts-Fourneaux, L-4362 Esch-sur-Alzette, Luxembourg. [6] Chemical, Biological and Environmental Engineering, Universitat Autònoma de Barcelona, Edifici Q, 08193 Bellaterra (Cerdanyola del Vallès), Spain. [7] University of Luxembourg, Rectorate, 2 Avenue de l'Université, L-4365 Esch-sur-Alzette, Luxembourg. ✉email: magdalena.calusinska@list.lu

In a world of finite biological resources, the agenda of the UN's Sustainable Development Goals challenges scientific community to develop transformative technologies that would enable the replacement of petroleum-based raw materials and energy with renewable bio-based feedstock. Plant biomass (lignocellulose), being the most abundant and renewable natural resource, could have many applications in different sectors[1]. *Miscanthus* sp. is a rhizomatous grass and owing to its adaptability to various environmental conditions, it shows high potential for sustainable production of lignocellulose over large geographical range[2]. Considering its important agronomic advantages (e.g. high biomass yield per hectare, reduced soil erosion and low fertiliser and pesticide requirements), it is suitable for different biorefinery value chains, including bioethanol, biogas, food additives, ingredients for cosmetics, biopharmaceuticals, bioplastics, biomaterials, organic fertilisers and animal feed[3]. Yet, due to the high recalcitrance (resistance of the cell wall components to enzymatic hydrolysis), its use is largely untapped[4].

In living organisms, enzymatic hydrolysis of lignocellulose is mainly driven by carbohydrate active enzymes (CAZymes[5]). Glycoside hydrolases (GHs) are the primary enzymes that cleave glycosidic linkages. Often, they are assisted by carbohydrate esterases (CEs), polysaccharide lyases (PLs) and other auxiliary enzymes (AAs). With its unique consortium of microorganisms, the termite gut is considered as the most efficient lignocellulose degrading system in nature[6]. Complete loss of gut cellulolytic flagellates in all evolutionary higher termites and acquisition of novel symbiotic bacteria led to improved lignocellulolytic strategies. It allowed for diet diversification from mainly wood-restricted to e.g. dry grass and other plant litter, herbivore dung and soil organic matter at different stages of humification[7]. Until now, most research has focused on endogenous endoglucanases of termites and cellulases from termite gut flagellates[8]. CAZymes from higher termite gut bacteria have only recently started receiving scientific attention[9].

Here, we investigated the higher termite gut system of *Cortaritermes* sp. (Nasutitermitinae subfamily) from French Guiana savannah. Using 16S rRNA gene amplicon profiling of termite gut bacteria, we investigated the adaptation of two termite colonies to *Miscanthus* diet under laboratory conditions. Through the de novo metagenomic (MG) and metatranscriptomic (MT) reconstructions, we assessed the distribution of activities within gut community, and we linked it to different bacteria, two main players being Spirochaetae and Fibrobacteres. Further analysis of gene expression profiles proved microbial functional plasticity (adaptation to changing environmental conditions through differential genes expression), and highlighted the abundance of gene transcripts involved in carbohydrate metabolism and transport. Analysis of the reconstructed community metagenome evidenced CAZyme clusters targeting two main components of *Miscanthus* biomass, namely cellulose and (arabino)xylan. Most of these clusters were allocated to the reconstructed metagenome assembled genomes (MAGs) of Fibrobacteres and Spirochaetae origin. The de novo reconstruction of the host epithelial gut transcriptome evidenced its contribution to lignocellulose degradation, and its adaptation to *Miscanthus* diet. Based on the characterisation of purified bacterial CAZymes, we verified the in silico predicted activities for many backbone-targeting (e.g. endocellulases and xylanases) and debranching enzymes (e.g. arabinofuranosidases). To finish, we discussed our findings in the context of enzymes application in the developing biorefinery sector.

## Results and discussion
### Structural alteration of gut microbiome under *Miscanthus* diet.
To examine enzymatic degradation of *Miscanthus* by the higher termite gut system, two laboratory-maintained colonies (nests LM1 and LM3) of *Cortaritermes intermedius* were fed exclusively with dried *Miscanthus* straw (Supplementary Figs. 1 and 2). This Nasutitermitinae genus is known to feed on grass tussocks in its natural habitat[10]. Alteration of the termite gut microbiome (here relative to bacterial communities in termite midgut and hindgut) was monitored at monthly basis during 9 months, by high-throughput sequencing the V6–V8 regions of 16S rRNA gene (Fig. 1a). Quality-trimmed reads were assembled into 678 operational taxonomic units (OTUs) assigned to 18 bacterial phyla. Spirochaetae and Fibrobacteres were the most dominant, as previously shown for plant fibre-feeding higher termites (e.g. ref. [11]; Supplementary Data 1). By assessing bacterial community structures in control samples (colonies feeding on grass tussocks in situ) and *Miscanthus*-fed microbiomes, we could observe radical changes. Species richness and diversity were significantly higher (HOMOVA $p < 0.001$) before *Miscanthus* diet was initiated, possibly reflecting an adaptive selection for the most efficient microbial degraders facing lower complexity of carbon sources in comparison to original diet (Fig. 1b). Further application of linear discriminant analysis (LDA) effect size (LEfSe; ref. [12]) to two termite colonies demonstrated that nearly 140 bacterial OTUs were significantly enriched in control microbiome, while roughly 13 were enriched in *Miscanthus*-fed microbiome (Supplementary Data 2). Out of the latter, six OTUs assigned to Fibrobacteres (mainly representing a genus exclusively containing termite Fibrobaceteres sequences) and Spirochaetae (associated with the termite Treponema cluster) were particularly abundant, and on average they accounted for 55.39% ± 3.8 of the *Miscanthus*-fed microbiome, in comparison to the 29.87% ± 1.8 average abundance in control microbiome. By analysing bacteria naturally associated with *Miscanthus* diet, we estimated the effect of immigration on the termite gut community as negligible (Supplementary Fig. 3). All together, these results demonstrated that diet change drives the development of microbial consortium in a unique manner, yet food-associated microbes cannot compete with highly specialised termite gut microbiota for a niche.

### Comparison of de novo metatranscriptomics and metagenomics.
The de novo MT was applied to nest LM1 and co-assembling reads from three samples (LM1_1 "control sample", LM1_2 and LM1_8, both representing "Miscanthus-fed microbiomes") yielded 603,579 open reading frames (ORFs), mainly representing partially reconstructed gene transcripts. The de novo MG reconstruction was also applied to colony LM1 at sampling point LM1_8 and it yielded 211,724 ORFs annotated on 64,347 contigs for the total assembly size of 177.5 million base pairs (Mbp). For both datasets, initial public database-dependent taxonomic classification of genes and gene transcripts pointed to the abundance of Firmicutes (Supplementary Fig. 4), what contrasted the results of community structure analysis. Subsequent binning of MG contigs and phylum-level annotation of the resulting bacterial bins allowed assigning correct taxonomic origin, confirming the metagenomic abundance of Spirochaetae and Fibrobacteres (Fig. 1c; Supplementary Fig. 5). Following mapping of RNA reads to the MG assembly (referred to as "RNA-seq" analysis), we could confirm transcriptional dominance of these two bacterial phyla as well. Incomplete public databases and extensive horizontal gene transfer were previously proposed as the origin of this misclassification[9].

Based on the classification of genes and transcripts to broad functional categories such as KEGG ontology profiles (KOs), congruency between the de novo MG and MT reconstructions was high (Supplementary Fig. 6). However, out of the de novo MT reconstructed gene transcripts of prokaryotic origin, only

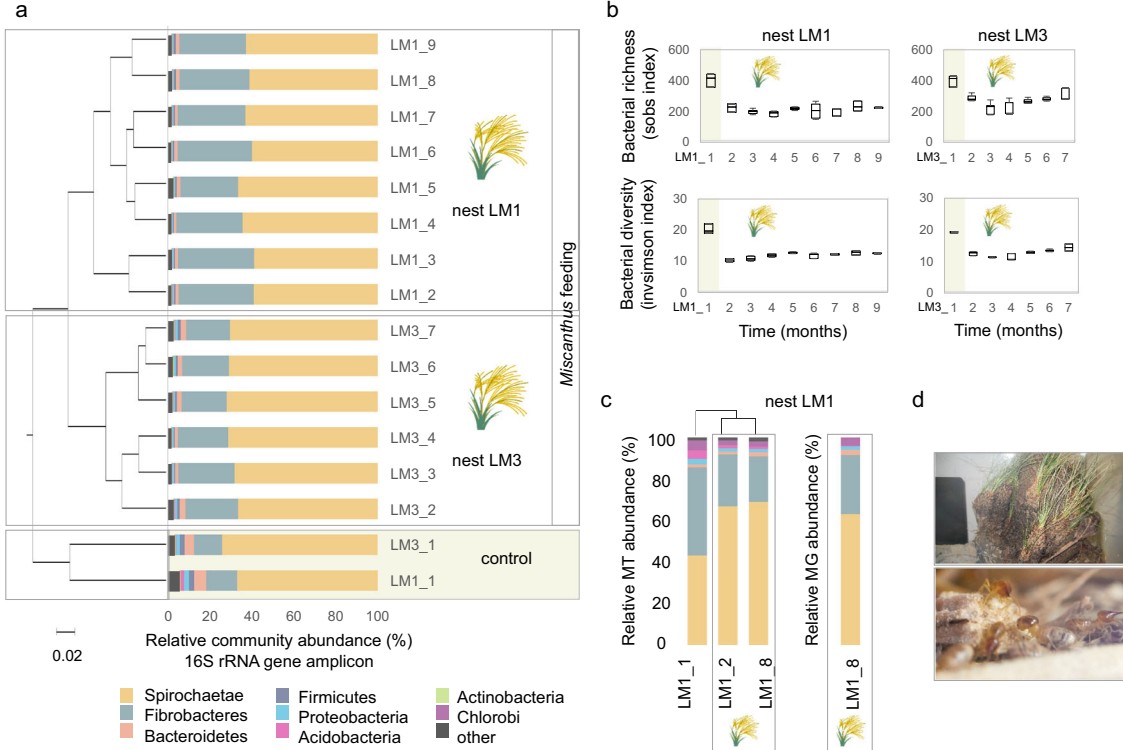

**Fig. 1 Structural composition of *Cortaritermes* sp. gut microbiome under *Miscanthus* sp. diet. a** Clustering of samples based on the calculated Bray−Curtis index and phylum level taxonomic assignment of sequencing reads from the 16S rRNA gene amplicon study. **b** Bacterial richness and diversity indices before (highlighted in yellow on sub-figures a and b) and under *Miscanthus*. Boxes represent the interquartile range and error bars show the 95% confidence intervals ($n = 3$). **c** Relative metatranscriptomic (MT) and metagenomic (MG) reads abundance assigned at the phylum level. Taxonomic gene and gene transcript assignments were inferred from the metagenomic contigs binning and phylum-level bin classification. **d** *Cortaritermes* sp. colony (top) and termite workers under the protection of soldiers while feeding on *Miscanthus* fibres in laboratory conditions (bottom).

37.8% showed significant similarity to the de novo MG genes at the protein level (blastp e-value $\leq 10^{-5}$), sharing on average 76.04% of amino acid identity (Supplementary Fig. 4). Coherent to our study, differences in functional gene profiles between MG and MT reconstructions have been previously underlined[13]. Even in the context of the termite gut, some authors highlighted the value of the de novo MT assembly in retrieving highly expressed genes[9,14].

**Genomic potential and transcriptional adaptation of gut microbes**. Aggregation of $68.9 \pm 1.8\%$ of the de novo reconstructed gene transcripts into clusters of orthologous genes (COGs) pointed at functional microbiome stability at the different stages of feeding campaign (Supplementary Data 3). Consistently with previous reports[11,14,15], cell motility and chemotaxis together with carbohydrate transport and metabolism were the two most highly expressed gene categories. Reconstruction of (nearly) complete metabolic modules was quite similar between Fibrobacteres and Spirochaetae. However, further comparative analysis using LEfSe (Fig. 2; Supplementary Data 4 and 5) identified several biologically informative features differentiating these two bacterial phyla. Both were capable of nitrogen fixation and glycogen synthesis, but the two pathways were enriched in Fibrobacteres. Expression of Amt ammonium transporters was highly up-regulated, and together with increased abundance of gene transcripts involved in urea transport and metabolism (restricted to Spirochaetae), it indicated nitrogen deficiency of a *Miscanthus*-fed termite colony. Both Spirochaetae and Fibrobacteres could also synthesise ten essential amino acids that animals cannot synthesise de novo. Even though nitrogen provisioning by bacterial symbionts is not employed by all

herbivorous insects, this strategy was proposed as a mechanism contributing to the success of termites[14] and herbivorous ants[16] in their marginal dietary niches. Importance of lignocellulose degradation under *Miscanthus* diet was evidenced by increased abundance of transcripts broadly assigned to cellulose and xylan processing KOs (Fig. 2; Supplementary Data 4). Multiple sugar ABC transporters were up-regulated in the Spirochaetae metatranscriptome, while they were nearly absent from the MG and MT reconstructions of Fibrobacteres origin. This observation could suggest the governance of exogenous carbohydrates uptake and utilisation by Spirochaetae.

**Diversity and abundance of termite gut microbial CAZymes**. The de novo MT reconstruction yielded over 2000 of manually curated transcripts assigned as CAZy-coding genes (*cazymes*). Out of these, 38.4% were further assigned to 55 GH families. The de novo MG reconstruction resulted in close to 7000 different *cazymes*, 43.6% of which were assigned to 86 GH families (Supplementary Figs. 6 and 7). Although there was a good correlation between the distributions of identified GHs to different GH families (Pearson $r$ 0.83), roughly 150 *cazymes* identified in the de novo MT were also reconstructed from the assembled metagenome. Novelty of reconstructed *cazymes* was evidenced through sequence comparison to NCBI refseq database, and a metagenomic dataset previously generated for *Nasutitermes* sp.[15] (Supplementary Fig. 4). In the latter case, average amino acids identity for the 943 query hits equalled $65.4 \pm 19.9\%$ (blastp, e-value $\leq 10^{-5}$), pointing to the diversity of microbial *cazymes* in guts of different termite species, even those phylogenetically closely related.

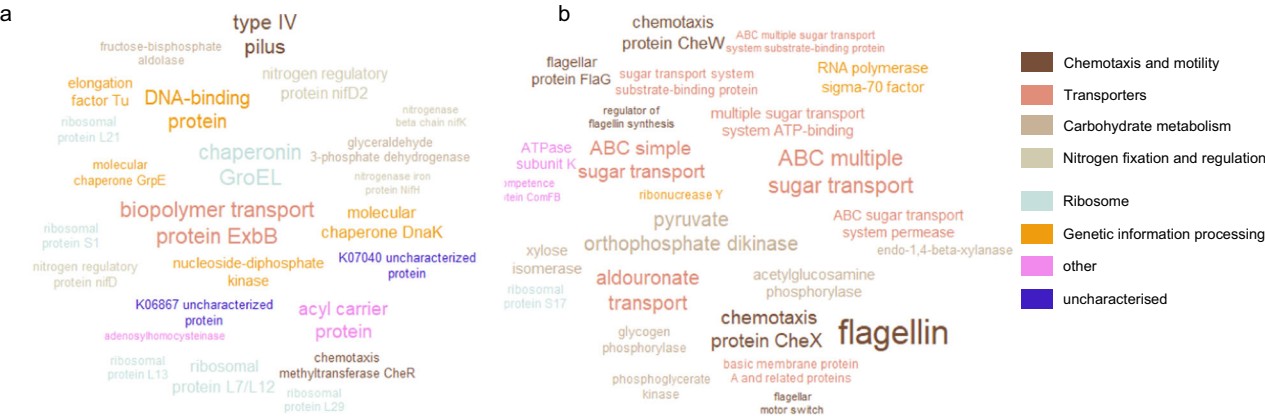

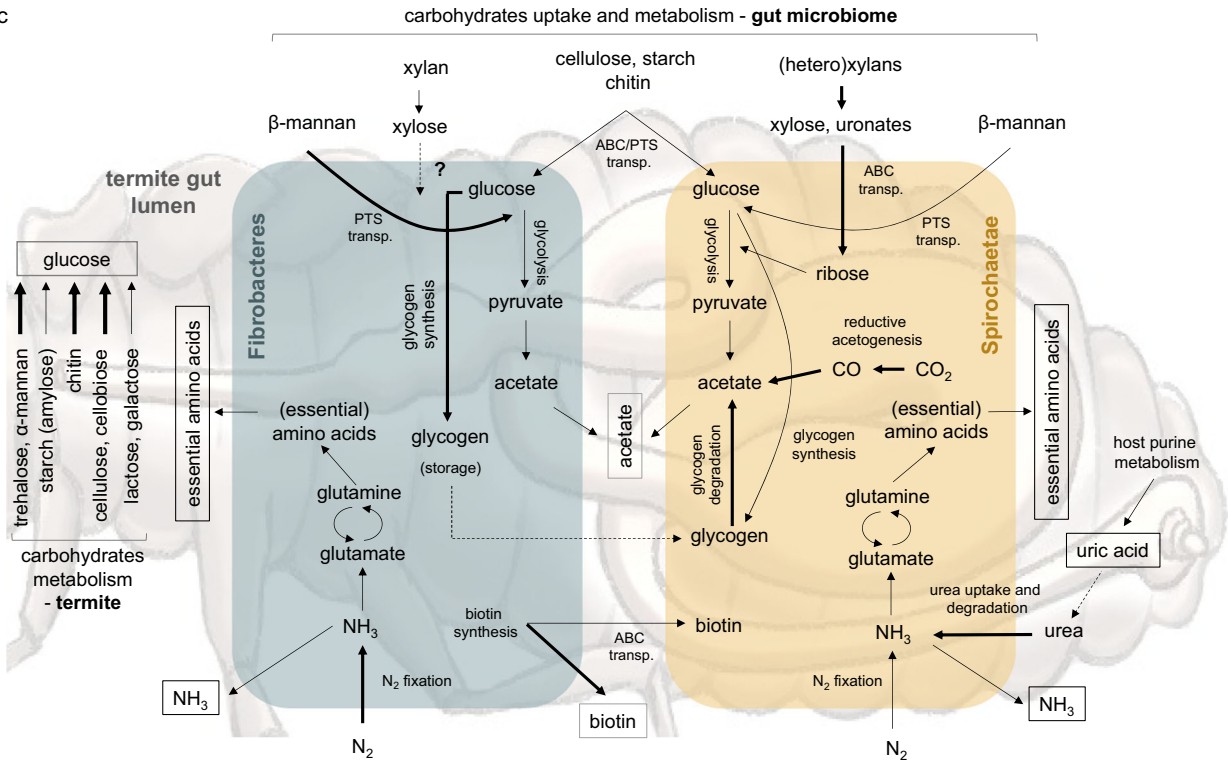

**Fig. 2 Functional characterisation of the termite gut system feeding on *Miscanthus* sp. a, b** Tag clouds of enriched (LefSe LDA > 2, *p* < 0.05) KOs reconstructed from the de novo metatranscriptomics for the termite gut Fibrobacteres (**a**) and Spirochaetae (**b**) at LM1_8. Top 25 most abundant KOs are displayed. Size of the text reflects transcriptomics abundance of a specific KO. **c** Simplified metabolic reconstruction, with a focus on carbohydrate metabolism, for the termite gut lignocellulolytic system. Hypothetical pathways are indicated with dashed lines. Metabolic pathways enriched in Fibrobacteres and Spirochaetae (metatranscriptomes) are indicated with bold lines. Metabolites putatively shared between gut bacteria and the host are indicated with square boxes.

Differential expression of specific *cazymes* at different stages of the feeding experiment suggested quick acclimation to new (laboratory) conditions, also reflecting adaptation of gut microbes to digest *Miscanthus* (Fig. 3). There were roughly 29.7% of common *cazymes* transcripts between LM1_8 and control LM1_1 sample, while over 55% were shared between LM1_8 and LM1_2 (both fed with *Miscanthus*). Along the experiment, GH5 (mainly subfamilies GH5_4 and GH5_2) was the most highly expressed family. Still, its cumulative expression nearly doubled under *Miscanthus* diet (Fig. 3b). Other abundant families included GH43, GH10 and GH11, all potentially involved in (hetero)xylan degradation. The latter was previously shown as largely expressed by the termite gut fibre-associated Spirochaetae[9]. Following manual curation, we removed three highly

abundant but only partially reconstructed GH11 gene transcripts, what reduced initial over-dominance of this CAZy family by 3.3 ± 0.9-fold (Supplementary Fig. 8). Similarly, only highly fragmented GH11 genes were recovered from the reconstructed metagenome. We hypothesise that closely related Spirochaetae strains contain highly similar GH11 genes, possibly shared by horizontal gene transfer; likewise, it was shown for e.g. human gut Bacteroidetes[17]. Their conserved nature may impede proper gene reconstruction from sequencing reads, similarly to other structurally conserved genes such as 16S rRNA or transposase, that typically do not reconstruct into larger genomic fragments[18]. GH45 was only present in Fibrobacteres and it was the second most expressed GH family under *Miscanthus* diet. According to the CAZY database, nearly 95% of proteins assigned to this family

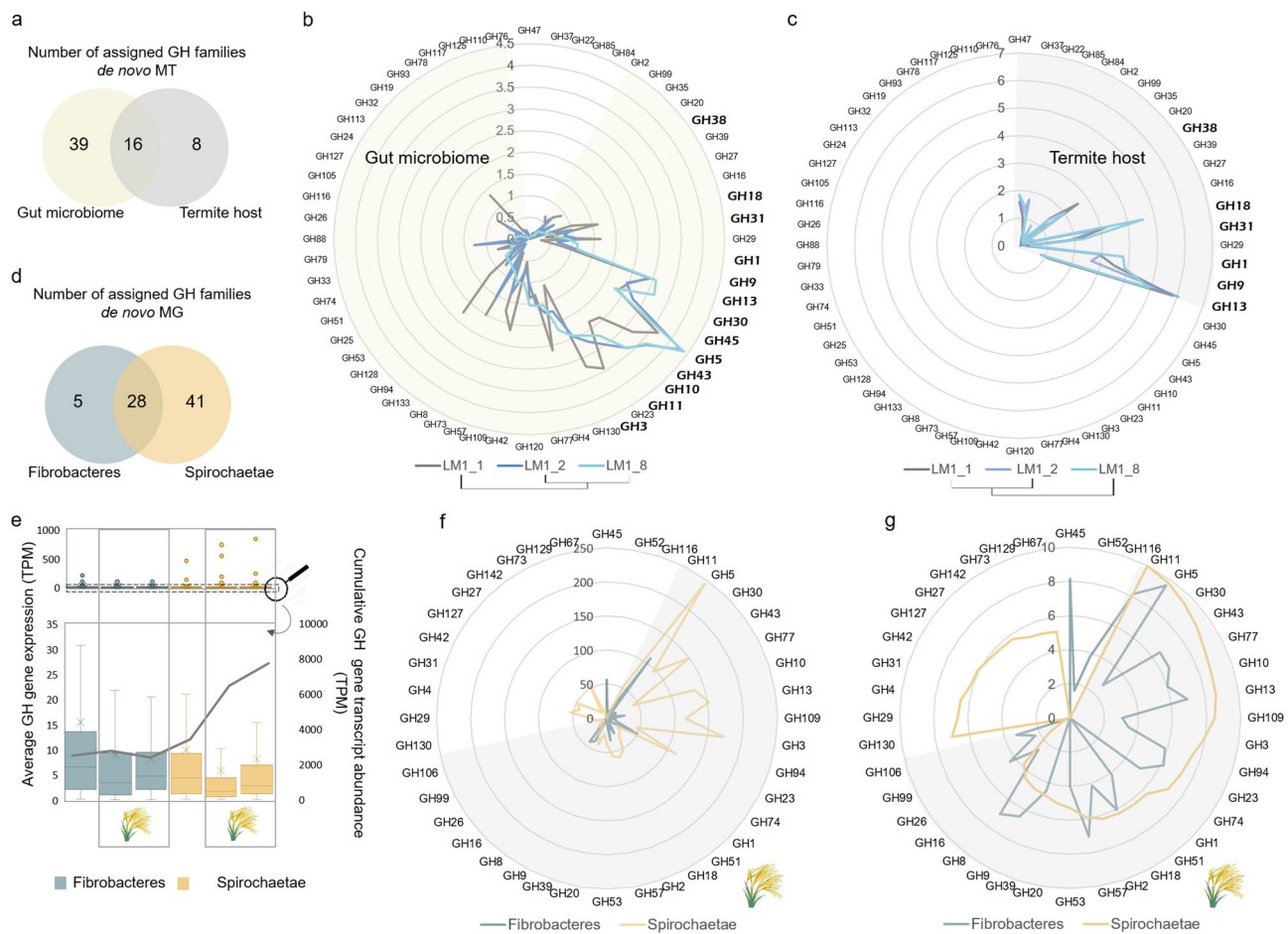

**Fig. 3 Carbohydrate active enzymes (CAZymes) reconstructed from metatranscriptomic (MT) and metagenomic (MG) reads for the termite gut system. a** Venn diagram showing the number of assigned glycoside hydrolase (GH) families for the termite gut microbiome and host gut epithelium. **b**, **c** Comparison of the gene expression profiles (de novo MT, log2 transformed) for gene transcripts assigned to the different GH families and at the different stages of the *Miscanthus* feeding experiment, for the gut microbiome (**b**) and the host gut epithelium (**c**). **d** Venn diagram showing the number of assigned GH families for Fibrobacteres and Spirochaetae, based on the de novo MG reconstruction. **e** Average CAZyme genes expression and cumulative gene expression at the different stages of the feeding experiment and analysed separately for Fibrobacteres and Spirochaetae. Lower panel is a zoom on the gene expression profiles with the outliers (highly expressed genes; in some cases representing only partially reconstructed genes) removed. Boxes represent the interquartile range and error bars show the 95% confidence intervals (n = number of transcripts annotated as glycoside hydrolases). **f**, **g** Number of genes (**f**) and cumulative abundance of the most abundant GH families (**g** RNA-seq log2 transformed) at the time point LM1_8 (the end of the *Miscanthus* feeding experiment), and visualised separately for Fibrobacteres and Spirochaetae. Transcripts abundance (**g**) is calculated based on the RNA mappings (RNA-seq) to the MG contigs. Shaded parts correspond to shared GH families.

are of eukaryotic origin (Supplementary Fig. 10), and show endocellulase activity. Previously, GH45 CAZymes were characterised from lower termite symbiotic protists[19]. Lytic polysaccharide monooxygenases (LPMOs) typically assigned to AA9 (fungal) and AA10 (predominantly bacterial enzymes) families[20] were neither present in the reconstructed metagenome nor metatranscriptome.

**Expression and activities of GHs from Fibrobacteres and Spirochaetae.** Based on the RNA-seq analysis, Fibrobacteres and Spirochaetae expressed respectively 47.9 ± 14.8% and 45.6 ± 18.5% of their *cazymes* genomic content when the termite was fed with *Miscanthus*. Total diversity of Spirochaetae *cazymes* was 2.3 ± 0.5-fold higher than for Fibrobacteres (Fig. 3d–g). Their cumulative transcriptional abundance was also higher for Spirochaetae; however, calculated average gene expression was slightly higher for Fibrobacteres. This observation was consistent across different GH families (Supplementary Fig. 11). As various GH families are characterised with broad functionalities, using

peptide-based functional annotation[21], we further assigned in silico specific functions (EC numbers) to 60.3 ± 2.2% of gene transcripts classified as GHs. In many cases, these predictions were experimentally validated (Supplementary Data 6). We confirmed β-glucosidase, endocellulase, endoxylanase and arabinofuranosidase activities of several Spirochaetae CAZymes. We also characterised active endoxylanases and endomannanases from Fibrobacteres.

Abundance of transcripts associated with endocellulase (EC:3.2.1.4) and endoxylanase (EC:3.2.1.8) increased under *Miscanthus* diet (for both Fibrobacteres and Spirochaetae), while those involved in chitin and starch (α-glucans) degradation decreased (Fig. 4; Supplementary Fig. 12). Endocellulase-assigned transcripts were nearly equally abundant between Fibrobacteres and Spirochaetae, while abundance and diversity of endoxylanases of Spirochaetae origin was much higher. Most of the assigned endocellulases were classified as GH5_4 enzymes (Supplementary Fig. 8.). Phylogenetic reconstruction comprising the previously characterised CAZymes from this family revealed the presence of multiple protein clusters separately grouping

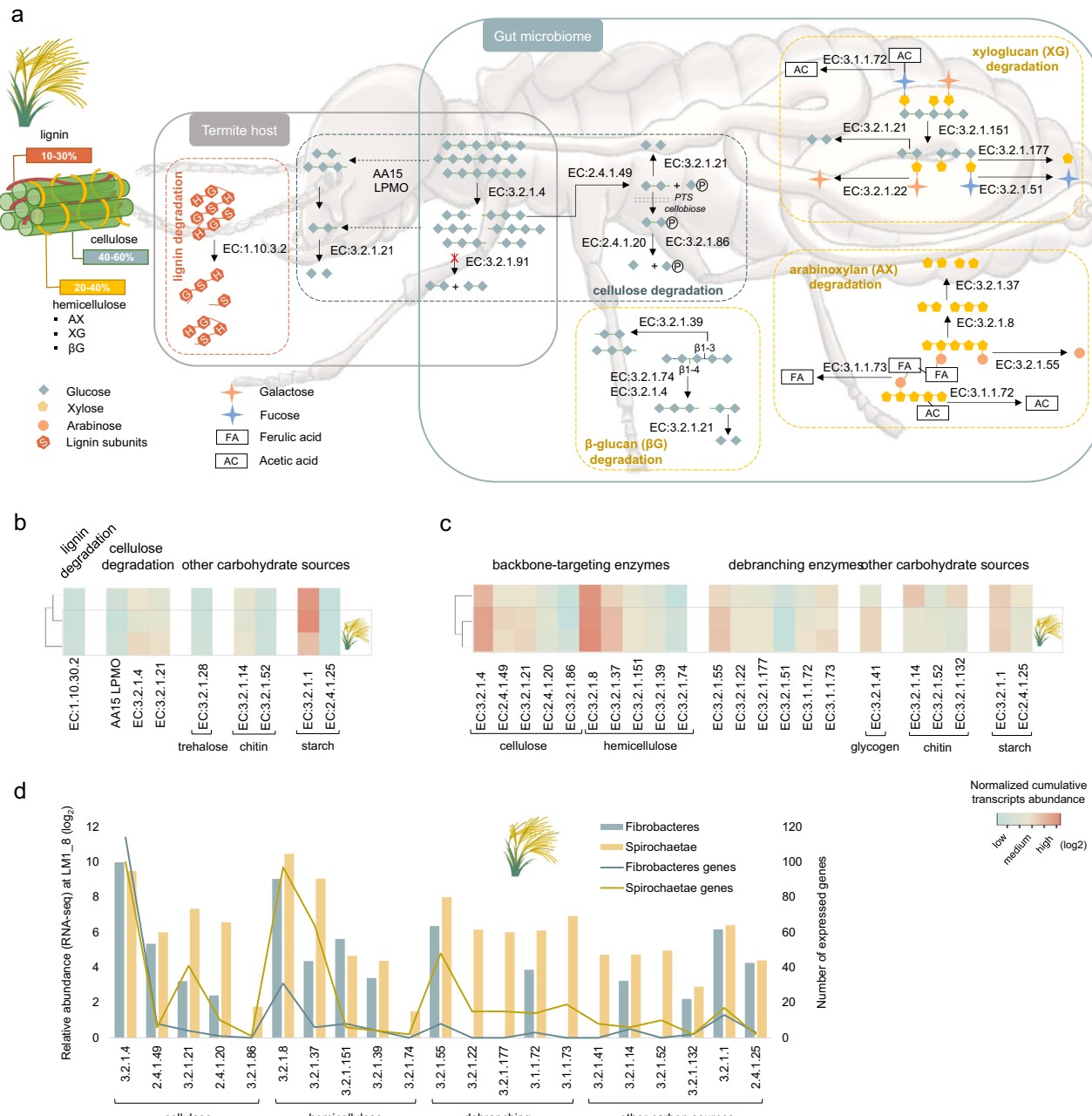

**Fig. 4 Characterisation of the termite gut lignocellulose degradation strategies. a** Simplified overview of enzymatic pathways involved in the degradation of main components of the *Miscanthus* biomass, based on enzymes (gene transcripts assigned an EC number) revealed in our study. Dashed lines indicate hypothetical pathways. Lignin subunits correspond to: p-hydroxyphenyl (H), guaiacyl (G), and syringyl (S). CAZymes gene expression profiles (de novo MT) at the different stages of the *Miscanthus* feeding experiment analysed for the termite gut epithelium (**b**) and the termite gut microbiome (**c**). Gene expression analyses were done separately for the termite gut epithelium and the gut microbiome; therefore, data on sub-figures b and c should only be compared within a single sub-figure. **d** Relative CAZymes gene transcripts abundance (RNA-seq) and gene numbers assigned to different enzymatic categories and analysed separately for Fibrobacteres and Spirochaetae for LM1_8 sample.

Spirochaetae and Fibrobacteres GHs (Fig. 5a). Concurrent inspection of reconstructed genomic fragments suggested the existence of different *cazymes* loci containing GH5_4 genes (Supplementary Fig. 13). Interestingly, CAZymes previously characterised to possess single enzymatic activity (mostly endocellulase and to a lower extent endoxylanase) grouped in upper part of the tree. Lower part of the tree mainly contained multi-functional enzymes (single enzyme simultaneously acting on cellulose and xylan). Suggested enzymatic multifunctionality was further confirmed for a selected GH5_4

CAZyme representing Spirochaetae cluster IX. Purified protein was shown to be an endocellulase acting on carboxymethylcellulose (CMC) and glucomannan (Fig. 5b). In addition, activity on xylan and arabinoxylan was confirmed. This gene was also one of the most highly expressed *cazymes* under *Miscanthus* diet, hypothesising the importance and interest for bacteria to express multi-functional enzymes. To our best knowledge, it represents the first GH5_4 CAZyme of Spirochaetae origin ever characterised, and first multi-functional enzyme of higher termite gut prokaryotic origin.

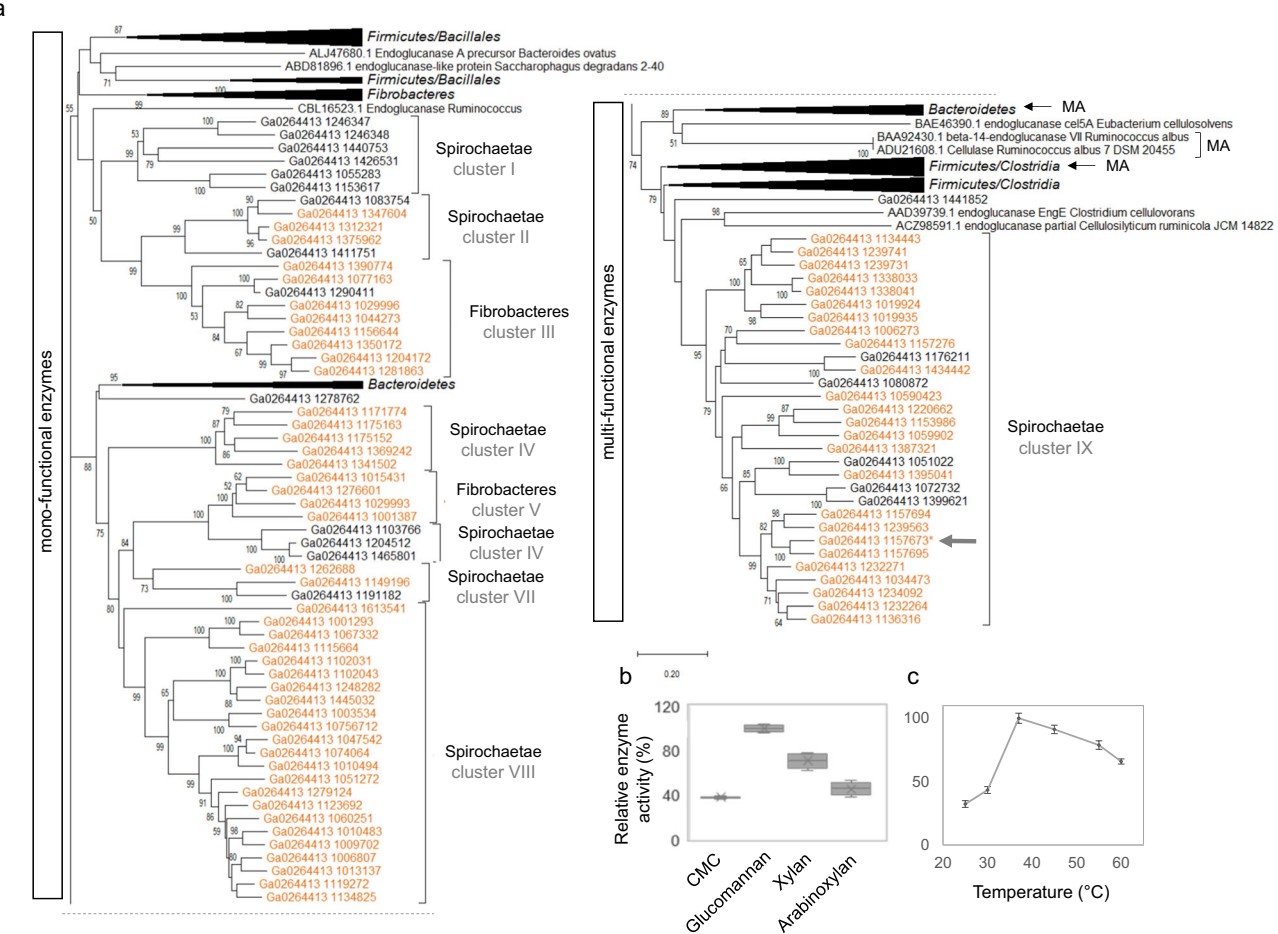

**Fig. 5 Characterisation of the GH5_4 family. a** Unrooted neighbour-joining tree containing the de novo reconstructed genes from MG study (genes expressed under *Miscanthus* diet are highlighted in orange on the tree). Tree was cut in two parts along the dashed line. All GH5_4 characterised proteins were retrieved from the CAZY database and included on the tree. Clusters indicated with an arrow and designated as "MA" contain known multi-functional enzymes. The percentage of replicate trees in which the associated sequences clustered together in the bootstrap test (500 replicates) are shown next to the branches. Final alignment involved 157 amino acid sequences. Protein from the *Spirochaetes* cluster IX indicated with a grey arrow was heterologously produced and characterised. **b** Activity profiles for the heterologously produced and purified protein tested against CMC, glucomannan (galactomannan was negative, not displayed on the graph), xylan and arabinoxylan. Boxes represent the interquartile range and error bars show the 95% confidence intervals (n = 4). **c** Optimal temperature was assessed for glucomannan substrate. Error bars represent the standard deviation of a dataset (n = 3).

**MAGs reconstruction and carbohydrate utilisation gene clusters.** Reconstruction of draft microbial genomes enriched in *Miscanthus*-fed termite gut microbiome resulted in 20 MAGs with completeness above 50% and contamination below 10%, including eight Fibrobacteres-assigned MAGs, six Spirochaetae (four of Treponema origin) and three novel MAGs representing Proteobacteria phylum (Supplementary Data 7 and Supplementary Fig. 14). Average nucleotide identity to MAGs from gut microbiomes of several higher termite species[22] equalled roughly 77.1 ± 1.8%, confirming the novelty of our MAGs. Frequency of *cazymes* was higher in the Treponema genomes and they were also enriched in GHs. Over 36% of annotated *cazymes* were aggregated into 1096 gene clusters containing more than one CAZy encoding gene. Similar gene clusters were recently discovered in gut microbiota of a wood-feeding termite *Globitermes brachycerastes*[23]. Putative cellulose-utilisation gene clusters were the most highly expressed in Fibrobacteres MAGs, suggesting its major contribution to cellulose degradation (Figs. 3 and 4). Endoxylanase-encoding genes were often co-localised with endocellulases (Supplementary Data 7); however, xylosidases were scarce, questioning the ability of Fibrobacteres to utilise complex xylans. *Cazymes* clusters targeting among others alpha glucans, (arabino)xylans (AX), beta glucans, cellulose, chitin, galactans, mannans and xyloglucans were evidenced in reconstructed Spirochaetae genomes. Largely complete AX-targeting clusters were assigned to MAG_17 and MAG_1, both representing Treponema. All of them were transcriptionally up-regulated under *Miscanthus* diet (Supplementary Fig. 15), what is consistent with high AX content of *Miscanthus* hemicellulose. Genes encoding for carbohydrate transporters (Supplementary Data 8) and in some cases even the two-component system response regulators and chemotaxis were adjacent to several *cazymes* clusters. Organisationally similar to polysaccharide utilisation loci (PULs) systems employed by Bacteroidetes[24], *cazymes* clusters reconstructed in our study more resemble the concept of Gram-positive PULs recently proposed by Sheridan et al.[25] in the context of human gut Firmicutes. Taking into account high sequence similarity of Spirochaeteae and Firmicutes *cazymes*, it is possible that whole *cazymes* clusters were acquired from Firmicutes in the course of evolution. Although not yet shown for Spirochaetae, Bacteroidetes PULs are often encoded within integrative and conjugative elements enabling their transfer among closely related species[17].

**Host functional gene expression profiles under Miscanthus diet.** Following taxonomic annotation, 16,416 gene transcripts of eukaryotic origin were classified to *Arthropoda*. Based on the presence of 271 conserved orthologous reference eukaryotic genes[26], we estimated the completeness of our termite gut epithelial transcriptome (relative to midgut and hindgut) at 89.5%, while the contamination with foreign mRNA was below 0.7%. Number of assembled ORFs was in line with the two published termite genomes, *Macrotermes natalensis* (16,140 protein-coding genes; ref. [27]) and *Zootermopsis nevadensis* (15,459 protein coding-genes; ref. [28]; Supplementary Data 9 and Supplementary Fig. 16). However, it was much lower than for another sequenced lower termite genome of *Cryptotermes secundus* (26,726 protein coding-genes; accession number PRJNA432597). Similar to gut microbiome, gene transcripts related to carbohydrate transport and metabolism were abundant in the host transcriptome, showing importance of carbohydrates metabolism to the termite lifestyle (Supplementary Data 3). Over 10,000 transcripts were assigned a KO number and a summary of complete and partially reconstructed metabolic pathways is provided in Supplementary Data 4. Except for a previously partially reconstructed wood-feeding *Nasutitermes takasagoensis* transcriptome with 10,910 detected transcripts, our reconstruction represents the most complete transcriptome of a higher termite representing Nasutitermitinae subfamily[29].

We further identified 170 CAZY gene transcripts assigned to four main classes (GH, CE, AA and GT) and associated CBMs. Glycoside hydrolases were encoded by 66 genes, and their diversity patterns were similar to those identified in other termites with sequenced genomes (Supplementary Data 9). They were assigned to 19 different GH families, out of which five were not represented in the gut microbiome (Fig. 3a–c). The highest transcriptional abundance was attributed to GH13 (typically assigned as alpha glucanases; Supplementary Fig. 17), and it slightly decreased only towards the end of *Miscanthus* campaign. Based on rather constant expression profiles of chitodextrinase, chitin utilisation by the host did not change significantly upon *Miscanthus* feeding (Supplementary Fig. 9). This indicates constant complementation of diet with nitrogen rich chitin, originating from either necrophagy and/or cannibalism, or fungi-colonised food stored in the nest, as previously proposed for other higher termite species[30]. Transcriptional abundance of endocellulases increased at later stages of *Miscanthus* feeding, suggesting a shift towards increased cellulose utilisation by host. In addition, a gene transcript sharing 54% of identity (at protein level) with a newly characterised cellulose- and chitin-targeting AA15 LPMO from *Thermobia domestica*[31] was also identified. This insect has a remarkable ability to digest crystalline cellulose without microbial assistance. Further blast analysis revealed a presence of homologous genes in the other termite genomes, including *Z. nevadensis, M. natalensis* and *C. secundus*, suggesting that next to certain eukaryotes (e.g. crustaceans, molluscs, chelicerates, algae, and oomycetes) termites might be able to oxidatively cleave glycosidic bonds. Similar observation was also recently stated[32].

Interestingly, a few gene transcript of eukaryotic origin were not assigned to Arthropoda, suggesting the presence of active small eukaryotes in the higher termite gut. One gene transcript was assigned to AA3 family and classified as putative cellobiose oxidoreductase (EC 1.1.99.18). This type of oxidase is involved in oxidative cellulose and lignin degradation in wood-decaying fungi[33]. In addition, two other transcripts assigned to GH45 family (putative endoglucanases) were present in the reconstructed transcriptome, and their expression slightly increased under *Miscanthus* diet. Homology search (blastp) revealed their closest similarity to *Anaeromyces contortus*, sp. nov. (Neocallimastigomycota), an anaerobic gut fungal species recently isolated from cow

and goat faeces[34]. In insects, GH45 *cazymes* were previously detected in the genomes of *Phytophaga* beetles[35] and until now they were not reported from sequenced termite genomes.

**Diet on Miscanthus: who does what?** *Miscanthus* biomass is mainly composed of cellulose (41.4 ± 2.9%), hemicelluloses (25.8 ± 5.2%) including arabinoxylans, xyloglucans, β-glucans, and lignin (21.4 ± 3.6%) and other trace components[4]. Based on the annotation of GH profiles and provided there is virtually no foreign nucleic acid contamination in the reconstructed termite transcriptome, termite on its own could digest amylose (starch), cellulose and/or cellobiose, lactose, galactose, chitin, mannan (α-mannan presumably contained in the fungal cell wall) and mannose, trehalose, other glycans (e.g. *N*-acylsphingosine) and bacterial cell wall components (Fig. 4a, b). Until now, hemicellulose degradation seems to be conducted by gut microbes, and no putative hemicellulolytic genes were recovered from the reconstructed termite transcriptome. For comparison, recent discovery of multi-functional GH9 cellulases in Phasmatodea suggested that some insects are capable to target xylan and xyloglucan in addition to cellulose[36]. The same study evidenced multi-functionality for two GH9 cellulases from *Mastotermes darwiniensis*, suggesting that some endogenous GH9 members in termites might also hydrolyse hemicellulose. Slightly increased transcriptional abundance of laccase-coding gene might indicate termite ability to target lignin. Beyond relatively high lignin content, recalcitrance of *Miscanthus* biomass is mainly enhanced by other features, in particular acetylation and esterification with ferulic acid (FA). While acetyl esterase activity can be deduced from Fibrobacteres and Spirochaetae metatranscriptomes, the ability to break ferulic linkages seems limited to Spirochaetae. Putative feruoyl esterase from CE1 family are contained within AX-targeting *cazymes* clusters in Spirochaetae MAGs, and were all up-regulated under *Miscanthus* diet (Supplementary Fig. 15).

Based on the diversity and expression patterns of GHs, different sugar transporters (mainly ABC and to a lesser extent PTS; Supplementary Data 8) and specific sugar isomerases, Spirochaetae are able to utilise a wider range of *Miscanthus*-derived sugars (including glucose, glucoronate, rhamnose, arabinose, mannose, xylose, ribose and fucose) than Fibrobacter*es* (mainly glucose, mannose and possibly ribose). Both bacterial phyla can target the backbone of cellulose, xylans and mannans (the latter is not abundant in *Miscanthus* biomass). Enrichment of Fibrobacteres metatranscriptome in endoglucanases (both targeting 1–3 β and 1–4 β glycosidic bonds as present in β-glucans and cellulose, respectively) shows its preference for carbohydrates with a glucose-unit backbone. Fibrobacteres also express endoxylanases, and we could confirm experimentally xylanase activity for one GH11 enzyme (Supplementary Data 6). However, hardly represented xylosidase-assigned gene transcripts and the absence of any xylose transporters and other known genes involved in xylose utilisation would question the ability of Fibrobacteres to utilise xylans. Co-localisation of many endoxylanases together with potential endocellulases in the reconstructed Fibrobacteres MAGs further suggest that termite gut Fibrobacteres mainly remove xylan polymers from *Miscanthus* fibres to better expose cellulose to the action of own endocellulases (Supplementary Fig. 13). By contrast, xylose isomerases and xylulose kinases were enriched in Spirochaetae metatranscriptome and both were highly expressed under *Miscanthus* diet. All reconstructed gene transcripts were assigned to Spirochaetae, and together with enrichment of endoxylanase transcripts, it confirms the ability of these bacteria to degrade xylans, as recently proposed by Tokuda et al.[9].

Based on the in silico prediction of enzyme sub-cellular localisations, most of the endoxylanases from Fibrobacteres and

Spirochaetae are exported outside the cell (Supplementary Data 10), suggesting initial degradation of xylan backbone in the extracellular space. Many Spirochaetae endocellulases are also putative extracellular enzymes or anchored to the outer membrane. In contrast, multiple Fibrobacteres endocellulases possibly lack signal peptide and are assumed to be localised in the cytoplasm. In general, as much as half of Fibrobacteres GHs are predicted to be localised in cytoplasm. At the same time, three times more GHs are exported outside the cell by Spirochaetae. This could indicate a rather selfish carbohydrates degradation strategy employed by termite Fibrobacteres, where cellulose fibres primarily detached by extracellular endocellulases are transported inside the cell for further breakdown. Selfish carbohydrate capture and degradation was previously proposed for Bacteroidetes in the rumen[37] and anaerobic digestion reactors[38]. By contrast, recent work done on a rumen isolate *Fibrobacter succinogenes* S85 indicates that enzymes involved in cellulose degradation are localised on the cell surface[39]. By maximising intracellular cellulose breakdown termite gut Fibrobacteres would avoid being in competition with much more abundant Spirochaetae. Enrichment of *exbB* (encoding for a biopolymer transport protein ExbB; Fig. 2a) gene transcripts in Fibrobacteres metatranscriptome would suggest possible cellulose/cellodextrin uptake through a mechanism similar to an experimentally demonstrated TonB-dependent transport of maltodextrins across outer membrane of *Caulobacter crescentus*[40], also discussed for F. *succinogenes*[39].

**Conclusions**. Retrieving lignocellulose-active enzymes from naturally evolved biomass-degrading systems, with the use of continuously improving high-throughput sequencing technologies, presents a promising strategy to identify new enzymes with potentially enhanced activities. Limited metatranscriptomic reports highlight high representation and overexpression of CAZymes in termite digestomes (e.g. refs. [9,14]). In higher termite guts, many lignocellulolytic steps are assisted by gut microorganisms (e.g. cellulose deconstruction), while some are exclusively attributed to hindgut bacteria (e.g. hemicellulose degradation). Cellulose degradation capacities of different lignocellulose degrading environments, including the termite gut system[41], have extensively been studied in the past. Decomposition of hemicellulose and general mobilisation of different lignocellulose components (breaking bonds between diverse plant polymers) have received comparably less scientific focus. Importantly, there is an increasing industrial interest in xylan-processing enzymes[42], regarding their application in biomass (wood) processing, pulp bio-bleaching, animal nutrition, food additives, etc.[43]

According to the recently established glycome profile of *Miscanthus* sp., next to glucose, xylose and arabinose are the two main cell wall monosaccharides, both originating from arabinoxylan fibres which are esther-cross-linked by ferulic acid[44]. Consequently, *cazymes* specifically targeting (feruloylated) arabinoxylan components were highly up-regulated under *Miscanthus* diet, making them potentially suitable candidates for industrial xylan-targeting applications. Many of these genes were combined in clusters with a set of complementary hydrolytic activities to degrade e.g. AXs. Nature-optimised synergy between enzymes of the same CAZyme cluster could further provide the basis to better define industrially relevant enzymatic cocktails. Specific lignocellulose fractions could be selectively targeted to deliver desired products, with potential effects being fully expectable and controllable[38]. It would allow for their fine-tuned degradation for a variety of applications, e.g. oligoarabinoxylans for food industry (prebiotics), lignin fibres for biomaterials, glucomannans as food additives, etc. Feruloyl esterases, by removing cross-links between polysaccharides and lignin help separating lignin from the rest of biomass, offering an alternative and/or complementation to currently applied industrial treatments[45]. In addition, ferulic acid and other hydroxycinnamic acids can have many applications in food and cosmetic industries due to their antioxidant properties etc.[46], thus further extending the application range of *Miscanthus* biomass.

Approach-wise, experimental design undertaken in this study represents the enrichment strategy where a nature-derived microbial inoculum is grown in liquid batch cultures. Here, a natural system of the termite gut was shown to progressively adapt yielding a consortium of microbes specialised in degradation of *Miscanthus* biomass. Integrative omics combined with protein characterisation provides a framework for better understanding of complex lignocellulose degradation by the higher termite gut system and paves a road towards its future bioprospecting.

## Methods

**Nest origin, laboratory maintenance and sampling**. Initially, three colonies of grass-feeding higher termites from *Nasutitermitinae* (nests: LM1, LM2 and LM3) were identified in January 2017 in tropical savannah in French Guiana, in proximity to Sinnamary town (radius of 5 km to GPS: N 05°24.195′ W 053°07.664′). Termite nests were transported to the laboratory where colonies were maintained in separate glass containers at 26 °C, 12 h light and 12 h dark, and 90% humidity conditions. Termite colonies were fed with dried *Miscanthus* grass winter harvest rich in recalcitrant lignocellulose[47], for a period of up to 9 months. Colony LM1_2 died after few months and was excluded from extended analysis. Mature worker-caste individuals were sampled in regular monthly time intervals before (sample taken before *Miscanthus* diet corresponds to "wild-microbiome") and following *Miscanthus* diet (samples correspond to "*Miscanthus*-adapted microbiome"). Termite specimens were cold immobilised, surface-cleaned with 80% ethanol and 1× phosphate-buffered saline and decapitated. Whole guts (here relative to the midgut and hindgut compartments) were dissected ($n \approx 30$ per replicate, minimum three replicates per sample) and preserved directly in liquid nitrogen. Additionally, for a sample selected for metagenomics analysis (LM1 time point 8 months; LM1_8), the hindgut luminal fluid was collected as previously described[11]. Samples were stored at −80 °C until further processing. Termite species were identified by morphology and by sequencing of the partial COII marker gene, as described before[48].

**Extraction of nucleic acids**. Total DNA and RNA were co-extracted from all samples using the AllPrep PowerViral DNA/RNA Kit (Qiagen) following the manufacturer's protocol. To assure the proper disruption of bacterial cell wall and termite gut epithelium cells, mechanical bead-beating step with 0.1 mm glass beads at 20 Hz for 2 min was introduced to complement the chemical lysis. The eluents were divided in two parts. First part was treated with 1 μL of 10 μg/mL RNase A (Sigma) for 30 min at room temperature. The second part was treated with TURBO DNA-free kit (Invitrogen) according to the manufacturer's protocol. The resulting pure DNA and RNA fractions were quality assessed using agarose gel electrophoresis and Bioanalyser RNA 6000 Pico Kit (Agilent). Nucleic acid concentration was quantified using Qubit dsDNA HS Assay and Qubit RNA HS Assay Kit (Invitrogen). DNA and RNA were stored at −20 °C and −80 °C, respectively.

**16S rRNA gene amplicon high-throughput sequencing and data analysis**. The bacterial 16S rRNA gene amplicon libraries were prepared using Illumina compatible approach as previously described[49]. Briefly, modified universal primers S-D-Bact-0909-a-S-18 (ACTCAAAKGAATTGACGG) and S-*-Univ-*-1392-a-A-15 (ACGGGCGGTGTGTRC, ref.[50]), and Nextera XT Index Kit V2 (Illumina) were used along with Q5 Hot Start High-Fidelity 2× Master Mix (New England Biolabs) in a two-step polymerase-chain reaction (PCR). In the first step selective amplification of the 484 bp long fragments of bacterial 16S rRNA gene V6–V8 region was performed. llumina-compatible adapters and barcodes were attached on the second step. Purified and equimolarly pooled libraries were sequenced along with PhiX control (Illumina) using MiSeq Reagent Kit V3-600 on in-house Illumina MiSeq Platform. Usearch v.7.0.1090_win64 software[51] was used for quality trimming, chimera check, singletons removal and assignment of the obtained sequences to OTUs at 97% similarity level. Taxonomic affiliation of the resulting OTUs was performed with SILVA database v.128 (ref.[52]). Downstream analyses were performed with mothur[53] and R environment[54]. Bacterial richness and diversity were calculated using respectively sobs and invsimpson indices. The dissimilarity of bacterial community structures was calculated using the Bray−Curtis index. OTUs differentially abundant between the wild- and *Miscanthus*-adapted microbiomes were assessed using the LEfSe approach[12].

**De novo metagenomics and data analysis**. Sample LM1_8 was selected for metagenomic sequencing in order to reconstruct genomes/larger genomic fragments of the dominant microbes in the *Miscanthus*-adapted microbiome. Prokaryotic DNA was enriched from the total hindgut DNA extract using NEBNext Microbiome DNA Enrichment Kit (New England BioLabs). Following sequencing, over 170 Mbp raw reads were quality trimmed in CLC Genomics Workbench v.9.5.2 (Qiagen), using a phred quality score of 20, minimum length of 50 and allowing no ambiguous nucleotides, resulting in close to 154 million quality-trimmed paired reads. Quality-trimmed reads were assembled using the CLC's de novo assembly algorithm in a mapping mode, using automatic bubble size and word size, minimum contig length of 1000, mismatch cost of 2, insertion cost of 3, deletion cost of 3, length fraction of 0.9, and similarity fraction of 0.95. The average contig abundance was calculated as DNA-RPKMs (reads per kilo base per million mapped reads). This type of normalisation allows for comparing contigs (genomic fragments) coverage (abundance) values, as it corrects differences in both sample sequencing depth and contig length. ORFs were searched and annotated using the default pipeline integrated in the IMG/MER[55]. Information about KEGGs and COGs assignment was retrieved based on the IMG/MER annotations. Metabolic pathways/modules were reconstructed using the tool integrated in the online version of the KEGG database[56]. Initially, contigs were binned using myCC[57] what resulted in their separation into 35 phylum-level bins of relatively high contamination. This approach was undertaken to correctly assign phylum-level taxonomy to the resulting contigs. Subsequently, the bin refinement module integrated in MetaWRAP was used to fine-tune the resulting bins (MAGs) to the species/strain levels[58]. The completeness and contamination of the generated MAGs were assessed with checkM[59]. Taxonomic affiliation was assessed with PhyloPhlan[60]. Similarity to the previously reconstructed MAGs was verified with the FastANI[61]. MAGs abundance within the reconstructed metagenome was calculated as average of contigs metagenomic abundance (DNA-RPKMs, see above) assigned to a specific MAG. Given the relatively high microbial diversity in the termite gut, only 54.6% of the resulting MG sequencing reads could map back to the reconstructed MG contigs, potentially mitigating the rate of functional gene discovery if solely relaying on the de novo MG reconstruction. Phylogenetic analyses were performed on MAFFT-aligned protein sequences[62] using MEGA X[63].

**De novo metatranscriptomics, host transcriptomics and data analysis**. For three selected samples (LM1-1, LM1-2 and LM1-8) the (meta)transcriptomic analysis was performed using an optimised approach described earlier[11]. Ribo-Zero Gold rRNA Removal Kit "Epidemiology" (Illumina) was used to enrich the sample for prokaryotic and eukaryotic mRNA. Enriched mRNA was purified using Agencourt RNAClean XP Kit and analysed with Bioanalyser RNA 6000 Pico Kit (Agilent). In continuation, SMARTer Stranded RNA-Seq Kit (Clontech) was used according to the manufacturer's instructions to prepare sequencing libraries. Final libraries were quantified with High Sensitivity DNA Kit (Agilent) and KAPA SYBR FAST Universal qPCR Kit. Libraries were pair-end sequenced at the Luxembourg Centre for Systems Biomedicine (University of Luxembourg) using Illumina NextSeq 500/550 High Output v2-300 Kit. Over 285 million raw reads were quality trimmed in CLC Genomics Workbench v.9.5.2, using a phred quality score of 20, minimum length of 50 and allowing no ambiguous nucleotides, resulting in close to 214 million quality-trimmed reads. Contaminating rRNA reads were removed using SortMeRNA 2.0 software[64]. The resulting non-rRNA reads were used to perform de novo (meta) transcriptome co-assembly using the CLC assembly algorithm in mapping mode with default parameters, except for minimum contig length of 200, length fraction of 0.90 and similarity fraction 0.95. As a result, nearly 2 million contigs were assembled. Obtained contigs were further submitted to IMG/MER for taxonomic and functional annotation[55]. Following the taxonomic assignment, 759,451 transcripts of putative prokaryotic origin were selected for further analysis. Initial IMG/MER taxonomic annotation resulted in over-representation of transcripts of putative Firmicutes origin (Supplementary Fig. 4). As nearly no Firmicutes OTUs were detected using the 16S rRNA gene amplicon sequencing, transcripts were re-annotated based on the de novo assembled metagenome and contig binning, resulting in re-classification of virtually all Firmicutes-assigned contigs to Fibrobacteres and Spirochaetae. To complement the study and to characterise potential contribution of the termite host to *Miscanthus* digestion, transcripts of eukaryotic origin and taxonomically assigned to *Insecta* (based on the IMG/MER annotation) were further evaluated for the completeness of the de novo reconstructed transcriptome with the BUSCO pipeline[26] and using the Eukaryota database (odb9). There were only two duplicated genes out of the total of 303 searched groups, suggesting that the level of possible contamination of non-*Insecta* origin was very low (below 0.7%).

For both the de novo assembled metatranscriptome and termite transcriptome, in order to determine the relative abundances of transcripts across studied samples, sequencing reads were mapped back to the annotated transcript sets, using the CLC "RNA-seq analysis" mode, with default parameters except for minimum similarity of 0.95 over 0.9 of the read length, both strands specificity and 1 maximum number of hits per read. The mapping results were represented as TPMs (transcripts per million), what directly resulted in normalised reads counts.

**Identification of *cazymes* and enzymes characterisation**. Genes (metagenomics) and gene transcripts (metatranscriptomics) encoding for microbial CAZymes were detected using dbCAN (dbCAN-fam-HMMs.txt.v6; ref. [65]) and CAZy database[66]. Using the threshold of e-value of <1e$^{-18}$ and coverage >0.35 recommended for prokaryotic CAZymes resulted in the removal of a high number of putative CAZymes; therefore, additional manual curation was performed to maximise the number of entries retained for further analysis. Additionally, gene transcripts outliers (very partially reconstructed gene fragments with average MT expression significantly exceeding the average expression of other genes assigned to the same group) were manually identified and removed as they were considered as chimeric (additional Blast search was launched in each case). Homology to peptide pattern (Hotpep) was used to assign an EC class to the identified CAZymes (ref. [21]). Subcellular localisation of CAZymes was predicted using BUSCA web[67]. To link the degradation of the different lignocellulose fractions and subsequent sugar utilisation, we looked for the presence of suitable sugar transporters and also specific sugar isomerases and kinases. Eukaryotic CAZymes, including for other sequenced termite genomes, were further searched with dbCAN2 and using dbCAN-fam-HMMs.txt.v8, with new AA families included.

Genes encoding for CAZymes of interest, selected based on their predicted activities and their expression profiles, were further PCR amplified (Veriti™ 96 wells Thermal cycler, Applied Biosystems, Foster City, USA) and the resulting PCR products were purified using a PCR purification kit (Qiagen, Hilden, Germany). If any signal peptide was predicted (using LipoP version 1.0, http://www.cbs.dtu.dk/services/LipoP/, ref. [68]), it was removed before cloning to enhance cytoplasmic protein production. Purified PCR products were cloned into the pET52b(+) plasmid and expressed in *E. coli* Rosetta (DE3) strain (Millipore Corporation, Billerica, MA, USA), as previously described[38]. Cells were harvested by centrifugation ($5000 \times g$, 4 °C, 15 min) and re-dissolved in a lysis buffer (50 mM NaH$_2$PO$_4$, 10 mM imidazole, 300 mM NaCl, pH8). Proteins were released by sonication, cell debris were removed by centrifuge ($16,000 \times g$, 4 °C, 15 min) and subsequent filtration step (13-mm syringe filter, 0.2-μm-pore-size). Affinity tag purification was achieved using a histidine tag located at the C terminus of a recombinant protein. NGC™ Medium-Pressure Liquid Chromatography system (Bio-Rad) equipped with a Hitrap™ column of 5 mL (Bio-Rad, Hercules, CA, USA) was used to purify produced proteins. A constant flow rate of 5 mL/min was applied. Initially, column was equilibrated with six column volumes (CV) of lysis buffer. Following equilibration, 290 mL of sample was injected and washed with three CVs of mixed buffer (97% of lysis buffer and 3% of elution buffer, the latter composed of 50 mM NaH$_2$PO$_4$, 250 mM imidazole, 300 mM NaCl, pH8). First step of elution was achieved using ten CVs of a linear gradient of mixed buffer (from 3% of elution buffer plus 97% of lysis buffer to 50% of elution buffer plus 50% of lysis buffer), followed by four CVs of 100% of elution buffer and finally one CV of 100% lysis buffer to detach the remaining protein. Fractions of two milliliters were collected during the washing and elution steps, and were analysed by SDS-PAGE. The release of 4-nitrophenol (PNP assay) and/or reducing sugar (RS assay) was used to determine the activity of recombinant proteins. Briefly, 50 μL of a purified protein solution was incubated with 50 μL of substrate (respectively, 4-nitrophenol derivatives were used for PNP assay and polysaccharides for RS assay) and 100 μL (PNP assay) or 25 μL (RS assay) of citrate phosphate buffer (pH7 0.1 M citric acid, 0.2 M dibasic sodium phosphate). The targeted substrates included carboxymethylcellulose (CMC), arabinoxylan, galactomannan, glucomannan and xylan. Enzymatic reaction was carried out at 37 °C during 1 hour (PNP assay) or 30 min (RS assay). The rate of release of 4-nitrophenol was instantly monitored at 405 nm using SPECORD 250 PLUS (Analytic Jena). The release of reducing sugars was determined following the Somogyi-Nelson method (refs. [69,70]). All assays were performed in triplicates.

**Statistics and reproducibility**. Whenever relevant biological or technical replicates were included in our study, this information is provided in specific sections of the "Methods" chapter. All statistical tests used are indicated and the reference is provided. Correlation was calculated using Pearson correlation coefficient.

**Reporting summary**. Further information on research design is available in the Nature Research Reporting Summary linked to this article.

## Data availability

The raw sequencing data used in this study were deposited in the SRA database under the following project numbers: PRJNA587606 (16S rRNA amplicon sequencing), PRJNA587423 (Metagenomics) and PRJNA587406 (Metatranscriptomics). OTU sequences were deposited in GenBank under project numbers PRJNA586754 and PRJNA434195. COII nucleotide sequences are available in GenBank under accession numbers MN803317−19. MG and MT assemblies and all other data underlying the findings of this study are available from the corresponding author upon reasonable request.

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

## Acknowledgements

This research was funded through FNR 2014 CORE project OPTILYS (Exploring the higher termite lignocellulolytic system to optimise the conversion of biomass into energy and useful platform molecules/C14/SR/8286517), and co-funded through the grant PDR T.0065.15 from the Belgian F.R.S.-FNRS. We are grateful to Philippe Cerdan, Régis Vigouroux and the staff of the Laboratoire Environnement HYDRECO of Petit Saut (EDF-CNEH) for logistic support during the field work.

## Author contributions

Experiments and molecular analyses were planned by M.C., M.M., M.B. and D.S.-D. and carried out by M.C., M.M. and M.B. with technical support from B.U. and D.K. M.C. and M.M. analysed the results. M.B. performed protein studies. Y.R., D.S.-D., X.G. helped with the collection of termites. R.H. and P.W. provided the support with Illumina NextSeq sequencing. P.Gawron and X.G. provided support with bioinformatics analysis. M.C., M.M. and M.B. wrote the manuscript. P.D., Y.R., D.S.-D., P.Gerin, P.F. participated in the planning and coordination of the study and in the manuscript correction. All authors read and approved the final manuscript.

## Competing interests

The authors declare no competing interests.
