## [Peer Review File · Communications Biology]

Reviewers' comments:

Reviewer #1 (Remarks to the Author):

This study comprehensively analyzed the microbial community, expressions of host and bacterial genes involved in lignocellulose degradation and metabolism in the gut of a higher termite *Cortaritermes* sp., which were imposed to *Miscanthus* diet. Sequencing of 16S rRNA genes revealed the dominance of Spirochaetae and Fibrobacteres members among those belonging to 18 bacterial phyla detected. Metagenomic and metatranscriptomic analyses indicated that the majority of cazyme genes or transcripts were derived from Spirochaetae and Fibrobacteres, which also have genes involved in nitrogen fixation or ammonium transporters to compensate for nitrogen deficiency. The origins of these genes were further confirmed with MAGs, while the enzymatic activities of selected gene products were demonstrated by heterologous expressions. The authors sequenced eukaryotic transcriptomes from the gut and suggested that the host has cellulase genes and the gut symbionts exclusively produce hemicellulases and also assist cellulose digestion. Especially, the observations strongly indicate that the members of Spirochaetae can digest and assimilate diverse polysaccharides including arabinoxylans with monofunctional and multifunctional enzymes, while the members of Fibrobacteres are more specialized in cellulose degradation and assimilation in a selfish manner. This study is potentially of great interest to represent novel findings on lignocellulose degradation in higher termites and their gut microbiota from a fundamental viewpoint.

However, the authors intended to focus on an applicable aspect of this digestive system to degradation of recalcitrant grass fibers. The discussion section primarily described only outlines of the results relevant to industrial interests, while most of findings, which are fundamentally of great importance, were not detailed. It distracted the value of this study, so that I greatly encourage the authors to revise this paper to add discussions of their findings in more detail from a viewpoint of basic biology and microbiology.

Another concern is unclear compositions of cellulose, hemicellulose, and lignin in grass tussocks in their natural habitats, although those of *Miscanthus* were documented in the text and Figure 4. In this context, it was ambiguous whether or not the observed changes of microbial compositions and upregulated or downregulated expressions of cazyme genes were in response to the dietary alterations.

Specific comments

L251-288: I was wondering whether fungal genes were contaminated in the eukaryotic transcriptome dataset of this study, which should have been assessed carefully. The authors described the presence of transcripts encoding GH45 members and an AA3 cellobiose dehydrogenase. Although these genes are frequently found in cellulolytic fungi, they were not detected in three termite genomes sequenced to date (Tokuda 2019. *Adv. Insect Physiol.* 57, 97-136).

L261-L262: Although not very relevant to lignocellulose digestion, some transcriptomic analysis of the gut in Nasutitermitinae were previously reported (Kumara et al., 2015. *J. Insect Physiol.* 78, 1-8; Kumara et al. 2016. *Appl. Entomol. Zool.* 51, 429-440).

L286-287: AA15 members are present in the genomes of not only *Zootermopsis* but also *Cryptotermes* and *Macrotermes*. However, it is not clear whether these members are involved in degradation of cellulose or chitin.

L292-L296: A possible occurrence of fungal mRNA contaminations may affect this spectrum of digestive substrates. In addition, multi-functionality of termite cellulases has been suggested (Shelomi

et al. 2020. *Insect Mol. Biol.* 29, 124-135).

L327-L330: Cellulolytic systems employed by *Fibrobacter* have been extensively proposed (Ranson-Jones et al. 2012. *Microbial Ecol.* 63, 267-281; Burnet et al. 2015. *PLoS One* 10, e0143809; Arntzen et al. 2017. *Environ. Microbiol.* 19, 2701-2714; Raut et al. 2019. *Sci. Rep.* 9, 16542). The results should be compared with these previous studies and discussed.

Fig. 3b,c,f,g, ad supplementary figure 12

These circular diagrams were hard to see. Some characters were overlapped or too small, while lines and axes were not very clear. These data should be shown in another way (e.g. heatmaps, bar graphs, etc.).

Figure 4

What do "G", "S", and "H" of lignin subunit indicate?

Since this study did not deal with the salivary glands, it should be noted that the digestive system of the host is only partially represented.

Reviewer #2 (Remarks to the Author):

The objective of the study was to characterize the overtime microbiome shifts in the higher termite gut to a grass diet, *Miscanthus* in efforts to identify the lignocellulose degrading enzyme produced by both the host and the microbial symbionts. The study was very interestingly conducted by feeding the wild populations on *Miscanthus* for a period of time to understand the microbial shifts in response to the diet fed. The authors found interesting results and identified the bacterial groups and their enriched enzymes to predict the importance of these enzymes to lignocellulose substrate degradation. It was impressive that the authors also characterized the substrate specificity of the enzymes thus utilizing and also validating the data generated from the metatranscriptomic study. Data analysis was thoroughly done and well interpreted.

However, it is not clear how the authors have determined that the insects adapted to *Miscanthus* diet. Please find some other minor editorial suggestions of the manuscript. Manuscript can be accepted for publication after these minor concerns have been addressed.

Specific comments:

Line 76: Replace "actors" with "players"

Figure S1: How was the adaptation of the insects to *Miscanthus* spp. In the laboratory assessed?

Line 96: Was the 16srRNA sequencing was performed every month for 9 months?

Line 103: Change "food associated bacteria" to bacteria naturally associated with *Miscanthus* diet used in this study"

Line 110: Why only LM1, 2 and 8 were chosen for the analysis?

Line 111: That is a lot of ORFs!

Line 126: Please mention if it is blastx algorithm

Line 138: Please elaborate on the lines "further in depth analysis of gene expression profiles evidenced adaption to the diet"

Line 150: Typically the carbohydrate groups are surrounded by the lignin polymers within lignocellulose. If the bacterial KO terms are enriched and expanded in xylan and cellulose processing, could it be possible for the host to contribute to partial lignocellulose degradation particularly in hydrolyzing the lignin polymer? Authors describe this in the schematic figure, but was not clearly mentioned in the discussion.

Line 160: Please enumerate Pearson's coefficient (r). R2 is not the pearson's coefficient. Please makes

these changes in the Figure S6 also.

Line 164: Also provide the average alignment length

Line 168: Briefly mention what the control treatment is for the study. Having the methods section at the end of the manuscript makes it difficult to follow the results. To make the manuscript an easier read please include important details of the methods in results section.

Line 207: It is not clear when the notation CAZymes and cazymes is used throughout the manuscript

Line 210: Mention the enzyme family name along with GH5 etc. or define the major enzymes present in this family briefly.

Line 229: Are these results observed in this study?

Line 234: Please provide reference

Line 236: Rephrase the sentence

Line 254: Mention how was the completeness assessed

Line 330: Or it could also mean that spirochates and the higher termites exhibit a mutual symbiosis as evidenced by the presence of secretory carbohydrate degrading enzymes and also ABC and other specific transporters for potential acquisition of the degraded food.

Line 453: Why were only these time points chosen? Also, does LM_1 correspond to one month old termites reared on Miscanthus? Or the samples collected before Miscanthus diet? What is the control of the experiment? It was mentioned in the results and figures, but not in methods.

Line 464 and 468: Replace M with million

Line 521: Delete "They"

Line 526: Include pH within the parenthesis

Line 529: Please mention the instrument used for the kinetic assays

Line 756: Figure 2a can be represented either in pie charts or bar graphs to show both the qualitative and quantitative distribution

Line 845: Mention the common name of GH-4 family i.e. Cellulases in the figure legend.

Dear Editor, Dear Reviewers

First of all, I would like to thank you for this fair and constructive revision. We believe that it greatly helped to improve the quality of our manuscript and to clarify some points that were not explicit from the beginning.

Please find below detailed answer to each of your comments.

On behalf of all the authors,

Magdalena Calusinska

Reviewers' comments:

Reviewer #1 (Remarks to the Author):

This study comprehensively analyzed the microbial community, expressions of host and bacterial genes involved in lignocellulose degradation and metabolism in the gut of a higher termite *Cortaritermes* sp., which were imposed to *Miscanthus* diet. Sequencing of 16S rRNA genes revealed the dominance of Spirochaetae and Fibrobacteres members among those belonging to 18 bacterial phyla detected. Metagenomic and metatranscriptomic analyses indicated that the majority of enzyme genes or transcripts were derived from Spirochaetae and Fibrobacteres, which also have genes involved in nitrogen fixation or ammonium transporters to compensate for nitrogen deficiency. The origins of these genes were further confirmed with MAGs, while the enzymatic activities of selected gene products were demonstrated by heterologous expressions. The authors sequenced eukaryotic transcriptomes from the gut and suggested that the host has cellulase genes and the gut symbionts exclusively produce hemicellulases and also assist cellulose digestion. Especially, the observations strongly indicate that the members of Spirochaetae can digest and assimilate diverse polysaccharides including arabinoxylans with monofunctional and multifunctional enzymes, while the members of Fibrobacteres are more specialized in cellulose degradation and assimilation in a selfish manner. This study is potentially of great interest to represent novel findings on lignocellulose degradation in higher termites and their gut microbiota from a fundamental viewpoint.

However, the authors intended to focus on an applicable aspect of this digestive system to degradation of recalcitrant grass fibers. The discussion section primarily described only outlines of the results relevant to industrial interests, while most of findings, which are fundamentally of great importance, were not detailed. It distracted the value of this study, so that I greatly encourage the authors to revise this paper to add discussions of their findings in more detail from a viewpoint of basic biology and microbiology.

>>The structure of our manuscript follows the one of multiple other studies published in Nature journals, where the results section often refers to the previously published articles, thus discussing

the new discoveries with previous findings. In total, we discussed our findings with around 40 previously published studies, at least three quarters of these refer to fundamental aspects. While, the discussion section itself is rather short and positions the main discoveries in a broader context. This structure minimizes overlap with results section, which in our opinion is very often the case of many published studies. Therefore, if the editor and the reviewer agree, we would like to keep the current format. However, we agree that some fundamental findings could have been better discussed in our study. Therefore, following the remark of the reviewer we added additional discussion lines, wherever we found it suitable or lacking. All the changes are highlighted in yellow in the revised manuscript.

Another concern is unclear compositions of cellulose, hemicellulose, and lignin in grass tussocks in their natural habitats, although those of *Miscanthus* were documented in the text and Figure 4. In this context, it was ambiguous whether or not the observed changes of microbial compositions and upregulated or downregulated expressions of enzyme genes were in response to the dietary alterations.

>>Colonies of *Cortaritermes* used in this study were collected in South American grasslands, where any woody vegetation was practically absent. Moreover, according to previous studies, including the one cited in the main manuscripts, *Cortaritermes intermedius* was previously shown to feed on grass tussocks. The vegetation covering the white sands savannas where the colonies were collected (GPM coordinates are given in the Methods section) are typically covered with short grass. Even though we did not precisely analyse the lignocellulose content and composition of these grass tussocks, most of the species present in this habitat are commelinoid monocots, with lignocellulose composition somehow similar to *Miscanthus*. Colonies grown in laboratory were maintained in boxes with sterilised sand and the only accessible food source was the *Miscanthus* biomass. They were fed this food from day two after collection. We assumed that they adapted to this food, because 1) colonies survived on *Miscanthus* up to 9 months when the experiment was carried (following this period the diet was no longer controlled, however this is out of the scope here); 2) by assessing the composition of their gut microbiomes at regular monthly basis and when compared to the control sample (taken *in situ*) we could observe a change (already in the first non-control analysed sample; the same OTUs were enriched in the two termite colonies feeding on *Miscanthus* and this information was better described in lines:102-108) in a community structure that was maintained over the period of 9 months (this can be observed on Fig1a); 3) the community structure was significantly different from the control sample, and it was further supported by 4) reduced bacterial diversity. The pattern of expression of specific *cazymes* was also different from the control sample and very similar between the samples LM1_2 (taken at the beginning of month 2, the first non-control sample) and LM1_8 (taken at the beginning of month 8), both corresponding to the *Miscanthus* diet. We assumed that there would be a change of *cazymes* expression patterns when the colony would be feeding on *Miscanthus*, and indeed the expression of genes coding for CAZymes specifically targeting *Miscanthus* lignocellulose components could have been observed; this is discussed in the text. However, we specifically targeted a species originally feeding on grasses, in order to facilitate its adaption to *Miscanthus* grass. Previously, we chose other *Nasutitermes* colonies collected in the forest (most probably feeding on much more woody biomass), and they did not adapt well to the *Miscanthus* diet and died after a few weeks of experiment (data not shown).

>>Taken together, we assume that the colonies did adapt to *Miscanthus*. However, the fact that they were taken from their natural habitat and further raised in laboratory might have also influenced the observed patterns to some extent (most probably this would not refer to CAZymes expression profiles).

Specific comments

L251-288: I was wondering whether fungal genes were contaminated in the eukaryotic transcriptome dataset of this study, which should have been assessed carefully. The authors described the presence of transcripts encoding GH45 members and an AA3 cellobiose dehydrogenase. Although these genes are frequently found in cellulolytic fungi, they were not detected in three termite genomes sequenced to date (Tokuda 2019. *Adv. Insect Physiol.* 57, 97-136).

>>The level of completeness and contamination of the reconstructed termite transcriptome was evaluated by BUSCO pipeline, as stated in the Methods section. The estimate of the completeness level was previously specified in the Results sections. Additionally, we provided the information about the contamination level, that based on the duplication of conserved genes was estimated to be below 0.7 %. Specific sentence was added in lines: 260-261. Based on this result we assume that the contamination with foreign mRNA was negligible in our termite transcriptome.

Concerning the GH45, indeed it was initially assigned as of arthropoda origin, based on the whole sequence homology. However, as we were surprised to see it in our termite transcriptome reconstruction, we did additional search and we could realise that most probably it of fungal origin (it might have also been transferred in the course of the evolution). Nevertheless, this gene was discovered in genomes of other insects, as specified in lines: 292-293. Due to the lack of significant hits and incorrect assignment of entries in public databases, sometimes, new discoveries should be taken with caution. For this reason, we clearly stated in the Results section, that the reconstructed GH45 might be of fungal origin. On the other hand, the other sequenced termite genomes represent either lower termites who rely on their cellulolytic protists to degrade cellulose, or higher termite living in symbiosis with fungi; again who helps them pre-digest biomass. Our transcriptome, represents higher termite of *Nasutitermitinae* family, a group that basically rely on their gut bacteria to help them digest lignocellulose. Therefore, it might be possible that they acquired in the course of their evolution additional cellulases (not detected in the other termite species, representing different feeding strategies). However, we do not want to go that far with our conclusions, having only one fungus-free higher termite transcriptome reconstructed. Although, there is another *Nasutitermes* transcriptomic study, as pointed by the reviewer (see below), it is far less complete than our reconstruction, and the published study is not discussed in the context of carbohydrate active enzymes.

>>Concerning, the AA3 enzymes, we run dbCAN on the proteins predicted for the other reconstructed termite genomes, and putative genes are also present in two of them (*Z. nevadensis* and *C.secundus*). Unless experimentally documented, we take this finding with caution and we highlighted it as hypothetical on the Fig. 4.

L261-L262: Although not very relevant to lignocellulose digestion, some transcriptomic analysis of the gut in *Nasutitermitinae* were previously reported (Kumara et al., 2015. *J. Insect Physiol.* 78, 1-8; Kumara et al. 2016. *Appl. Entomol. Zool.* 51, 429-440).

>>This transcriptomics study was somehow overlooked by us, thank you for pointing it out. Additional reference was added (the most recent one as it corresponds to the more complete transcriptome reconstruction than the first one), and the sentence was corrected to acknowledge

this previous study accordingly; lines: 270-272.

L286-287: AA15 members are present in the genomes of not only *Zootermopsis* but also *Cryptotermes* and *Macrotermes*. However, it is not clear whether these members are involved in degradation of cellulose or chitin.

>>Sentence was corrected to include the information on cellulose and chitin as target substrates; line 294. According to our blast analysis, no homologous gene (blastp done at protein level) was detected in *M. natalensis* genome. Additional reference was also added; line 299.

L292-L296: A possible occurrence of fungal mRNA contaminations may affect this spectrum of digestive substrates. In addition, multi-functionality of termite cellulases has been suggested (Shelomi et al. 2020. *Insect Mol. Biol.* 29, 124-135).

>>Regarding the number of reconstructed termite gene transcripts that closely corresponds to the predicted number of ORFs in two other sequenced termite genomes (*Z. nevadensis* and *M. natalensis*, the number was much higher in the case of *C. secundus*; additional information was added to the main text, lines: 264-265), similar GH profiles and the level of possible contamination with foreign nucleic acids that was below 0.7%, we assume that the provided information about the substrate spectrum is rather correct. Of course zero contamination cannot be stated at 100%, therefore we rephrased the sentence in lines: 303-308.

>>Concerning the second remark related to multi-functional termite cellulases, unless we are mistaken, this feature of multifunctional xylanolytic cellulases was shown to be unique to Phasmatodea insects only, and it was not detected for termite GH9 cellulases (Shelomi et al. 2020. *Insect Mol. Biol.* 29, 124-135).

L327-L330: Cellulolytic systems employed by *Fibrobacter* have been extensively proposed (Ranson-Jones et al. 2012. *Microbial Ecol.* 63, 267-281; Burnet et al. 2015. *PLoS One* 10, e0143809; Arntzen et al. 2017. *Environ. Microbiol.* 19, 2701-2714; Raut et al. 2019. *Sci. Rep.* 9, 16542). The results should be compared with these previous studies and discussed.

>>Additional discussion, including some of the recent studies was added; lines: 344-355. Not all of the proposed works could have been cited in our manuscript due to the limit imposed to the reference list.

Fig. 3b,c,f,g, ad supplementary figure 12

These circular diagrams were hard to see. Some characters were overlapped or too small, while lines and axes were not very clear. These data should be shown in another way (e.g. heatmaps, bar graphs, etc.).

>>We agree with the reviewer that some characters were small or even overlapping, that is why GH families broadly discussed in our manuscript were highlighted in bold and written using larger font in the figure 3. We would like to keep the current representation as we think it nicely presents the trends and highlight the similarity of GH profiles between the samples corresponding to the *Miscanthus* feeding. Additionally, by zooming the electronic version of the publication, all the details can be better seen. Axes are explained in figure legends if not clearly stated on the figure.

Figure 4

What do “G”, “S”, and “H” of lignin subunit indicate?

Since this study did not deal with the salivary glands, it should be noted that the digestive system of the host is only partially represented.

>>The explanation related to the lignin subunits was added to the figure legend. It was also clearly stated in the manuscript that the termite transcriptome corresponds to the midgut and hindgut compartments; line 260.

Reviewer #2 (Remarks to the Author):

The objective of the study was to characterize the overtime microbiome shifts in the higher termite gut to a grass diet, *Miscanthus* in efforts to identify the lignocellulose degrading enzyme produced by both the host and the microbial symbionts. The study was very interestingly conducted by feeding the wild populations on *Miscanthus* for a period of time to understand the microbial shifts in response to the diet fed. The authors found interesting results and identified the bacterial groups and their enriched enzymes to predict the importance of these enzymes to lignocellulose substrate degradation. It was impressive that the authors also characterized the substrate specificity of the enzymes thus utilizing and also validating the data generated from the metatranscriptomic study. Data analysis was thoroughly done and well interpreted.

However, it is not clear how the authors have determined that the insects adapted to *Miscanthus* diet. Please find some other minor editorial suggestions of the manuscript. Manuscript can be accepted for publication after these minor concerns have been addressed.

Specific comments:

Line 76: Replace “actors” with “players”

>Done

Figure S1: How was the adaptation of the insects to *Miscanthus* spp. In the laboratory assessed?

>>The adaptation of termite colonies to *Miscanthus* diet was assessed based on the change of bacterial community structures before (termite feeding natural diet, sample was collected *in situ* and preserved for further analysis) and after the introduction of *Miscanthus* to the diet. For both termite colonies, the same bacterial OTUs were enriched when fed *Miscanthus*. Even if the bacterial community structure was continuously evolving, the major change occurred just after the diet was introduced, this can be well seen on the Bray-Curtis tree in Fig. 1A. This observation is stated in lines: 96-108. Some additional discussion was added through the manuscript, all changes are indicated in yellow.

>>Please refer also to the answer to the reviewer 1 for further explanation.

Line 96: Was the 16srRNA sequencing was performed every month for 9 months?

>>Yes, the sentence was changed accordingly; lines: 92. The first sampling was done in situ (day 0), while all the other samplings were done with intervals of one month.

Line 103: Change “food associated bacteria” to bacteria naturally associated with miscanthus diet used in this study”

>>Sentence corrected.

Line 110: Why only LM1, 2 and 8 were chosen for the analysis?

>>Initially, we chose these three samples for metatranscriptomics sequencing. While LM1_1 represents the *in situ* taken sample (control), LM1_2 and LM1_8 refer to the *Miscanthus*-fed colony. The latter samples were taken at the beginning (one month after the introduction of *Miscanthus* diet) and towards the end of *Miscanthus* diet. Based on the results, we could observe that the adaptation occurred quickly after the diet was introduced (LM1_2) and was maintained towards the end of the experiment (LM1_8). Although, there was a continuous evolution of the gene expression profiles, the two samples corresponding to the *Miscanthus*-fed colony largely resemble each other in terms of the gene expression profiles, and are largely different from the control sample. For this reason, we assumed that the expression profiles of termite gut bacteria fed *Miscanthus* diet were similar at the different time points, and thus we prioritised complementary metagenomics sequencing rather than going for additional metatranscriptomics samples. Optimally, one would like to sequence and analyse all time points. However, the budget is never unlimited, and a good compromise between delivering good quality scientific results and the cost needs to be taken into account. Particularly, gene cloning and further protein purification and activity assays were the most time-consuming steps. However, we realised that it was important to complement our sequencing-based findings with enzymatic data. We truly believe that with our experimental design we did not compromise the importance of the outcomes of this study.

Line 111: That is a lot of ORFs!

>>The community in the termite gut is relatively diverse, and taken the applied sequencing depth, many ORFs were recovered.

Line 126: Please mention if it is blastx algorithm

>>The comparison was done at the protein level. Reconstructed genes (MG) and gene transcripts (MT) were translated to proteins using Prodigal and the comparison was done with blastp. The sentence was changed accordingly to avoid confusion; lines: 131-132.

Line 138: Please elaborate on the lines “further in depth analysis of gene expression profiles evidenced adaption to the diet”

>>Comparative analysis of enriched KO categories within the two main bacteria populations in the gut when the termite was fed with *Miscanthus* was done using LEfSe, the sentence was changed accordingly; lines: 143-144.

Line 150: Typically the carbohydrate groups are surrounded by the lignin polymers within

lignocellulose. If the bacterial KO terms are enriched and expanded in xylan and cellulose processing, could it be possible for the host to contribute to partial lignocellulose degradation particularly in hydrolyzing the lignin polymer? Authors describe this in the schematic figure, but was not clearly mentioned in the discussion.

>>Presence of putative lignin-targeting gene transcripts in the reconstructed termite transcriptome and their putative involvement to lignin degradation were discussed though the manuscript at two occasions; lines: 287, 311.

Line 160: Please enumerate Pearson's coefficient (r). R2 is not the Pearson's coefficient. Please make these changes in the Figure S6 also.

>>Corrected.

Line 164: Also provide the average alignment length

>>It was blastp comparison, all against all. The sentence was changed accordingly for clarity.

Line 168: Briefly mention what the control treatment is for the study. Having the methods section at the end of the manuscript makes it difficult to follow the results. To make the manuscript an easier read please include important details of the methods in results section.

>>Control sample refers to the termite feeding on grass tussocks in its natural habitat. Sample was taken *in situ*; the sentence was changed accordingly, lines: 174-175.

Line 207: It is not clear when the notation CAZymes and cazymes is used throughout the manuscript

>>"CAZymes" corresponds to carbohydrate active enzymes (protein level), and the notion was explained the first time it was used, lines: 57-58. Written as "cazymes" we refer to CAZymes-coding genes (nucleotide level), and the term was explained the first time used, line 161. We adapted this nomenclature following many other previously published studies related to CAZymes.

Line 210: Mention the enzyme family name along with GH5 etc. or define the major enzymes present in this family briefly.

>>For most of the GH families, it is difficult to designate a specific name, since in many cases several enzymatic activities are assigned to a single GH family. For this reason, to avoid confusion we prefer to stick to the GH family number only. We tentatively propose specific designation, like endoglucanase, xylanase *etc*, when a specific EC number could have been attributed to a protein with HotPep. Even though, as shown for one GH5_4 expressed and characterised enzyme, a multi-activity could have been demonstrated. That is why we prefer to stay prudent and avoid using common names when referring to specific GH families.

Line 229: Are these results observed in this study?

>>Yes, these results are represented in Supplementary fig. 14.

Line 234: Please provide reference

>>Reference to Supplementary Table 7 was added.

Line 236: Rephrase the sentence

>>The sentence was rephrased, lines: 242-243.

Line 254: Mention how was the completeness assessed

>>The completeness and contamination (this information was added in the revised manuscript), were estimated using BUSCO. The reference to the original study is given, and the description of the method is present in the Methods section; lines: 260-261; 499-501.

Line 330: Or it could also mean that spirochates and the higher termites exhibit a mutual symbiosis as evidenced by the presence of secretory carbohydrate degrading enzymes and also ABC and other specific transporters for potential acquisition of the degraded food.

>>This is a very interesting point raised by the reviewer. We believe that this would have to be further investigated.

Line 453: Why were only these time points chosen? Also, does LM_1 correspond to one month old termites reared on Miscanthus? Or the samples collected before Miscanthus diet? What is the control of the experiment? It was mentioned in the results and figures, but not in methods.

>>Please refer to your previous remark "Line 110: Why only LM1, 2 and 8 were chosen for the analysis?" concerning the chosen experimental design. Control sample LM1_1 was taken in situ (day 0) and the other samples were taken at the beginning of each consecutive month. This information has clarified in the manuscript.

Line 464 and 468: Replace M with million

>>Corrected along the manuscript.

Line 521: Delete "They"

>>Corrected.

Line 526: Include pH within the parenthesis

>>Corrected.

Line 529: Please mention the instrument used for the kinetic assays

>>This information was added.

Line 756: Figure 2a can be represented either in pie charts or bar graphs to show both the qualitative and quantitative distribution

>>We would like to keep it in its current version, as in our opinion it better emphasizes the enrichment of specific gene categories that are discussed in the text. The quantitative aspect is there and it is visualised with the size of the word. Thanks to the remark of the reviewer, we realised that it was not stated in the figure legend. This information was added; line: 809.

Line 845: Mention the common name of GH-4 family i.e. Cellulases in the figure legend.

>>Knowing that the different enzymatic activities can be attributed to a single GH family, we prefer to stay prudent with associating common names to GH families. Especially, in the case of GH5_4 we show that an enzyme can act as endoglucanase and endoxylanase, therefore we prefer to avoid attributing the name “cellulases” to this family.

Reviewers' comments:

Reviewer #1 (Remarks to the Author):

I read the revised paper and the authors' rebuttals. The manuscript has been revised appropriately to some extent, while I feel it still needs a couple of minor modifications.

I understand that the authors would like to keep the current format for the Discussion section. However, a large fraction of discussions has already included in the Results section, and the current Discussion section simply described some perspectives for industrial applications and conclusion rather than essential discussions of the obtained results. If Editor feels appropriate, my suggestion is to combine the Results and Discussion sections or to change the heading of the final paragraphs to 'Conclusion' instead of 'Discussion' as employed by previous papers in Communications Biology (e.g. Article number 87 published in 2020).

Although I do not oppose to your view that the termites adapted to *Miscanthus grass*, they might have been influenced by the laboratory conditions. Especially, one cannot exclude the possibility that the hindgut bacterial compositions were more or less influenced by the environmental alterations (from the open air to the laboratory). In this context, you should be careful about descriptions on the gut microbiomes with *Miscanthus* diet, which might have some impact on repertoires of enzyme genes, unless otherwise you have an appropriate control such as termites fed on the original grass tussocks under the same laboratory conditions.

In the rebuttal letter, the authors mentioned that AA3 members in other termites were retrieved by dbCAN, but AA15 members were searched using BLASTp, which failed to detect a putative gene in *Macrotermes*. Such a gene retrieval should have been conducted in a consistent manner. If you employ dbCAN2 (v8) with the genome of *M. natalensis*, you will find that three genes (i.e. Mnat_04422, Mnat_07964, and Mnat_14519) are affiliated with AA15 (ref. 34). One of them (i.e. Mnat_04422) was also detectable with Hotpep.

As I mentioned in my original comments, AA3 cellobiose dehydrogenases (i.e. AA3_1) are exclusively of fungal origin. However, AA3 glucose oxidases (i.e. AA3_2) are also found in insects (ref. 34). As indicated in Supplementary Table 9, termite genomes encode multiple AA3 members. Was the cellobiose oxidoreductase (was it determined by Hotpep?), which was mentioned in L285-L286, affiliated with AA3_1? If it was AA3_1 transcript with very low abundance, I still feel it could be a contamination from fungal mRNA. If it belongs to AA3_2, you should clearly mention it (but it was unlikely that Hotpep defined it as cellobiose dehydrogenase/oxidoreductase).

In the rebuttal letter and Supplementary Table 9, it appeared that the authors failed to find AA3 members from the genome of *M. natalensis*, but the genome of *M. natalensis* actually contains 19 genes belonging to AA3_2 in the genome based on my retrieval with dbCAN2 v8 (and 17 genes with Hotpep). Thus, I wondered if the database search was appropriately conducted.

The authors rebuttal against my point on multifunctional cellulases was that bifunctional xylanase/cellulase enzymes are restricted to Phasmatodea. This is true, but the sentence (L308-310) documented that hemicellulose degradation is 'exclusively' conducted by gut microbes. Shelomi et al. (2020) mentioned that two GH9 members of *Mastotermes darwiniensis* (i.e. MaDa2 and MaDa3) were bifunctional cellulase/xyloglucanases. Xyloglucan is also the major hemicellulosic component. Thus, without experimental data, the possibility of some endogenous GH9 members in termites as multifunctional cellulase/hemicellulase cannot be ruled out. Apart from GH9, since the authors assumed a possible involvement of the endogenous enzymes in mannan degradation (L306-

L307), it is probably overstatement that hemicellulose degradation is exclusively carried out by bacteria.

Reviewer #2 (Remarks to the Author):

Effort of the authors to include suggested changes to improve the quality and representation of the manuscript is appreciated. Especially including the suggested details in the methods section has clarified a lot of confusion in understanding the manuscript. The revised manuscript meets the expectations of communication biology journal and thus can be accepted for publication.

Dear Editor, dear Reviewer

Please find below our answers to your comments. Thank you for your positive criticism that significantly helped us to improve our manuscript. If the reviewer 1 would like to reveal his or her identity, please feel free to contact me by email. I would be happy to discuss with you our other termite studies. I find your remarks very constructive.

Reviewers' comments:

Reviewer #1 (Remarks to the Author):

I read the revised paper and the authors' rebuttals. The manuscript has been revised appropriately to some extent, while I feel it still needs a couple of minor modifications.

I understand that the authors would like to keep the current format for the Discussion section. However, a large fraction of discussions has already included in the Results section, and the current Discussion section simply described some perspectives for industrial applications and conclusion rather than essential discussions of the obtained results. If Editor feels appropriate, my suggestion is to combine the Results and Discussion sections or to change the heading of the final paragraphs to 'Conclusion' instead of 'Discussion' as employed by previous papers in Communications Biology (e.g. Article number 87 published in 2020).

> Following the request of reviewer and with the agreement of editor, we would like to rename the section "Results" to "Results and discussion", while changing the last paragraph into "Conclusion and perspectives".

Although I do not oppose to your view that the termites adapted to *Miscanthus grass*, they might have been influenced by the laboratory conditions. Especially, one cannot exclude the possibility that the hindgut bacterial compositions were more or less influenced by the environmental alterations (from the open air to the laboratory). In this context, you should be careful about descriptions on the gut microbiomes with *Miscanthus* diet, which might have some impact on repertoires of enzyme genes, unless otherwise you have an appropriate control such as termites fed on the original grass tussocks under the same laboratory conditions.

>Our study design included control samples that were taken in natural habitat while the colony was feeding its usual grass tussocks diet. We see your point of view, however our experimental design was different and now it is impossible to include a control colony fed natural diet under laboratory conditions, without redoing the whole experiment from the beginning. We have discussed the point of having a control colony fed with grass tussocks in laboratory conditions, with some other colleagues who are termite specialists, and indeed some of them share your point of view. While the others agree with us, that in any case it would have been extremely complicated to subtract the data resulting from the metatranscriptomics analysis of the control from *Miscanthus* test colony. The reason for this is that no two termite colonies are identical in terms of their gut microbiome composition, and on the way they would adapt to the laboratory conditions. You can already see it on the Fig1, where the 16S gene amplicon sequencing was applied to two different termite gut microbiomes, over the time. Even though the control samples (taken in situ) were quite similar,

under laboratory conditions and *Miscanthus* food, each gut microbiome evolved a bit differentially. In fact, in this study we took five colonies and only three and ultimately two adapted well and were retained in this study.

The best way to perform the experiment having a control sample as suggested by you, would be to take a significant amount of termite workers and keep them outside the nest (in specially adapted Petri dishes, as done in some other studies), while feeding one batch with control diet and the other with *Miscanthus*. To justify our study design, we aimed at long-term experiment showing an adaptation to *Miscanthus*, which we think would not be possible outside the nest. Initially we were not sure how fast the microbiome would need to adapt to the diet. However, we could observe that the gene expression patterns were quite similar between the samples taken at the beginning of the 2nd and 8th months (both feeding on *Miscanthus*). Moreover, the expression patterns of CAZymes were matching nicely the main lignocellulose components present in *Miscanthus*. That is why, we still assume that the colony adapted to feed on *Miscanthus*, even though the impact of laboratory conditions could be not excluded here. Here you are completely right. Therefore, we agree with you, that we should probably not unambiguously state that the colony was adapted to *Miscanthus*, because at the same time it was surely influenced by the laboratory conditions as well. For this reason, we reshaped several sentences across the manuscript, in most of the cases replacing “*Miscanthus*-adapted” to “fed with *Miscanthus*” under laboratory conditions. All the changes are highlighted in yellow. We hope this effort and our justification is now acceptable by you.

In the rebuttal letter, the authors mentioned that AA3 members in other termites were retrieved by dbCAN, but AA15 members were searched using BLASTp, which failed to detect a putative gene in *Macrotermes*. Such a gene retrieval should have been conducted in a consistent manner. If you employ dbCAN2 (v8) with the genome of *M. natalensis*, you will find that three genes (i.e. Mnat_04422, Mnat_07964, and Mnat_14519) are affiliated with AA15 (ref. 34). One of them (i.e. Mnat_04422) was also detectable with Hotpep.

> Our initial CAZymes search was done with dbCAN and v6 database. At the time the bioinformatics treatment of data was done, dbCAN and v8 database were not yet available. However, following the reviewer’s comment, this search was repeated with dbCAN2 and v8 database, where AA15 family is already contained, thus avoiding supplementary BLASTp search. Additional Table 9 was updated accordingly. Also, specific paragraphs in the main text were rewritten accordingly (please see all the changes highlighted in yellow). However, authors could not download genomic sequence of *M. natalensis* as it was no longer available in NCBI database (only mitochondrial genome was available on March 21st, see attached print screen). Therefore, for *M. natalensis* CAZymes content as listed in Poulsen et al., 2014 was used in Supplementary Table 9. As a result, AA15 was not listed, as it was not detected in original paper. We understand the reviewer’s suggestion, however we could not redo the search to reveal the presence of AA15 in that termite genome.

#	Organism Name	Organism Groups	Strain	BioSample	BioProje	Size(Mb)	GC%	Type	Replicons	CDS	Release Date
1	Macrotermes annandalei	Eukaryota/Animals/Insects		PRJNA378001	0.01495	33.01		mitochondrion	NC_034078.1 KV224516.1	13	05-Mar-2017
2	Macrotermes barneyi	Eukaryota/Animals/Insects		PRJNA175021	0.01594	33.14		mitochondrion	NC_018599.1 JX050221.1	13	12-Sep-2012
3	Macrotermes carbonarius	Eukaryota/Animals/Insects		PRJNA378096	0.014853	34.33		mitochondrion	NC_034046.1 KV224449.1	13	05-Mar-2017
4	Macrotermes fageiger	Eukaryota/Animals/Insects		PRJNA37810X	0.01495	34.01		mitochondrion	NC_034050.1 KV224469.1	13	05-Mar-2017
5	Macrotermes gresus	Eukaryota/Animals/Insects		PRJNA378956	0.01495	34.06		mitochondrion	NC_034110.1 KV224607.1	13	05-Mar-2017
6	Macrotermes malaccensis	Eukaryota/Animals/Insects		PRJNA378902	0.014949	33.94		mitochondrion	NC_034030.1 KV224417.1	13	05-Mar-2017
7	Macrotermes muelleri	Eukaryota/Animals/Insects		PRJNA378920	0.01495	35.45		mitochondrion	NC_034127.1 KV224669.1	13	05-Mar-2017
8	Macrotermes natalensis	Eukaryota/Animals/Insects		PRJNA267311	0.016325	34.38		mitochondrion	NC_025522.1 KX405637.1	13	14-Nov-2014
9	Macrotermes subhyalinus	Eukaryota/Animals/Insects		PRJNA172291	0.016351	34.44		mitochondrion	NC_018128.1 JX144937.1	13	10-Jul-2012
10	Macrotermes vthialatus	Eukaryota/Animals/Insects		PRJNA377965	0.01495	34.70		mitochondrion	NC_034054.1 KV224472.1	13	05-Mar-2017
11	Macrotermes yunnanensis	Eukaryota/Animals/Insects		PRJNA383201	0.015965	33.30		mitochondrion	NC_034288.1 KU900578.1	13	17-Apr-2017

As I mentioned in my original comments, AA3 cellobiose dehydrogenases (i.e. AA3_1) are exclusively of fungal origin. However, AA3 glucose oxidases (i.e. AA3_2) are also found in insects (ref. 34). As indicated in Supplementary Table 9, termite genomes encode multiple AA3 members. Was the cellobiose oxidoreductase (was it determined by Hotpep?), which was mentioned in L285-L286, affiliated with AA3_1? If it was AA3_1 transcript with very low abundance, I still feel it could be a contamination from fungal mRNA. If it belongs to AA3_2, you should clearly mention it (but it was unlikely that Hotpep defined it as cellobiose dehydrogenase/oxidoreductase).

> Putative cellobiose dehydrogenase was reclassified of fungal origin and the whole paragraph was adapted accordingly. Additional dbCAN2 search was done and all the data, including figures and tables were adjusted accordingly, as explained above. To double check for fungal contamination the dataset relative to Arthropoda was reassigned taxonomically, and the search for KO was repeated with the updated database. Additional table 4 was updated accordingly.

In the rebuttal letter and Supplementary Table 9, it appeared that the authors failed to find AA3 members from the genome of *M. natalensis*, but the genome of *M. natalensis* actually contains 19 genes belonging to AA3_2 in the genome based on my retrieval with dbCAN2 v8 (and 17 genes with Hotpep). Thus, I wondered if the database search was appropriately conducted.

> We did a mistake in the description of this supplementary file. In fact, CAZymes of *M. natalensis* origin were taken from original publication by Poulsen et al., 2014. The remaining termite genomes were search for CAZymes with dbCAN and CAZY database dbCAN-fam-HMMs.txt.v6. Obviously there are new CAZymes families in V8, that were not present in V6, for example AA15. For this reason, additional BLAST analysis was conducted to verify the presence of this family in termites. At the time when our bioinformatics analysis was conducted V8 was not yet available. Following the advice of reviewer, we decided to repeat CAZymes search in all sequenced termite genomes with dbCAN2 and using dbCAN-fam-HMMs.txt.v8. Corresponding text and figures and tables were corrected accordingly. However, as described above, we could not access the genome of *M. natalensis*, because it is no longer available in NCBI genome database. If the reviewer has it and would like to

share it with us, we could repeat the dbCAN2 analysis. Otherwise, we will keep the CAZymes as reported by Poulsen et al., 2014, for a comparison in our supplementary table 9.

The authors rebuttal against my point on multifunctional cellulases was that bifunctional xylanase/cellulase enzymes are restricted to Phasmatodea. This is true, but the sentence (L308-310) documented that hemicellulose degradation is 'exclusively' conducted by gut microbes. Shelomi et al. (2020) mentioned that two GH9 members of (i.e. MaDa2 and MaDa3) were bifunctional cellulase/xyloglucanases. Xyloglucan is also the major hemicellulosic component. Thus, without experimental data, the possibility of some endogenous GH9 members in termites as multifunctional cellulase/hemicellulase cannot be ruled out. Apart from GH9, since the authors assumed a possible involvement of the endogenous enzymes in mannan degradation (L306-L307), it is probably overstatement that hemicellulose degradation is exclusively carried out by bacteria.

> Thank you for this remark. Indeed, you are right. We updated our text accordingly and included reference to Shelomi et al. 2020. We also removed the sentence stating that the termite itself could target hemicellulose mannan, as in fact only alpha mannanases could have been identified with our tools.

Reviewer #2 (Remarks to the Author):

Effort of the authors to include suggested changes to improve the quality and representation of the manuscript is appreciated. Especially including the suggested details in the methods section has clarified a lot of confusion in understanding the manuscript. The revised manuscript meets the expectations of communication biology journal and thus can be accepted for publication.

REVIEWERS' COMMENTS:

Reviewer #1 (Remarks to the Author):

I hope all the authors stay healthy under the COVID-19 outbreaks.

Thank you for your patience to address my criticisms. I mostly understand your rebuttals and feel the manuscript has been greatly improved. Finally, I would like you to ask for a minor but essential revision regarding the CAZy repertoires in the genome of *Macrotermes natalensis*, which would warrant the accuracy of this paper.

Thanks to your detailed explanations in the rebuttal letter, I recognized that you have been in trouble to find the genome sequence of *M. natalensis* from NCBI database. In the publication by Poulsen et al. (2014), they mentioned that the assemblies are available in GigaScience Database, while only raw reads have been deposited in NCBI (SRA database). Thus, please visit the following URL, in which you will find the fasta file of all coding nucleotide and deduced amino acid sequences in the genome of *M. natalensis* (dataset 100057).

<http://dx.doi.org/10.5524/100055>

With these sequences, I think you will be able to quickly amend the relevant information in your manuscript.

Minor remark:

P13L316: It would be better to rephrase "some endogenous GH9 members in termites might also hydrolyze hemicellulose".

I don't have any further comments, but please let me know if you need to discuss this or another study any further. Here, I reveal my identity.

Prof. Gaku TOKUDA

Tropical Biosphere Research Center, University of the Ryukyus, Japan
tokuda@comb.u-ryukyu.ac.jp

Dear Editor, dear Reviewer

Please find below our answers to your comments. We would like to thank Prof. Tokuda for providing us the link to the genome sequence of *M. natalensis* and all his advices that helped us to significantly improve the quality of our study.

Reviewers' comments:

Reviewer #1 (Remarks to the Author):

I hope all the authors stay healthy under the COVID-19 outbreaks.

Thank you for your patience to address my criticisms. I mostly understand your rebuttals and feel the manuscript has been greatly improved. Finally, I would like you to ask for a minor but essential revision regarding the CAZy repertoires in the genome of *Macrotermes natalensis*, which would warrant the accuracy of this paper.

Thanks to your detailed explanations in the rebuttal letter, I recognized that you have been in trouble to find the genome sequence of *M. natalensis* from NCBI database. In the publication by Poulsen et al. (2014), they mentioned that the assemblies are available in GigaScience Database, while only raw reads have been deposited in NCBI (SRA database). Thus, please visit the following URL, in which you will find the fasta file of all coding nucleotide and deduced amino acid sequences in the genome of *M. natalensis* (dataset 100057).

<http://dx.doi.org/10.5524/100055>

With these sequences, I think you will be able to quickly amend the relevant information in your manuscript.

>Thank you for the accession link to the genome. We repeated the search for CAZymes using dbCAN2 and v8 database. Supplementary Table 9 was updated. All changes were also highlighted in yellow in the main text.

Minor remark:

P13L316: It would be better to rephrase “some endogenous GH9 members in termites might also hydrolyze hemicellulose”.

>This sentence was rephrased as suggested

I don't have any further comments, but please let me know if you need to discuss this or another study any further. Here, I reveal my identity.

Prof. Gaku TOKUDA

Tropical Biosphere Research Center, University of the Ryukyus, Japan
tokuda@comb.u-ryukyu.ac.jp